# Insights into the inhibition of protospacer integration via direct interaction between Cas2 and AcrVA5

Mingfang Bi[1], Wenjing Su[1], Jiafu Li[1] & Xiaobing Mo [1,2] ✉

Spacer acquisition step in CRISPR-Cas system involves the recognition and subsequent integration of protospacer by the Cas1-Cas2 complex in CRISPR-Cas systems. Here we report an anti-CRISPR protein, AcrVA5, and reveal the mechanisms by which it strongly inhibits protospacer integration. Our biochemical data shows that the integration by Cas1-Cas2 was abrogated in the presence of AcrVA5. AcrVA5 exhibits low binding affinity towards Cas2 and acetylates Cas2 at Lys[55] on the binding interface of the Cas2 and AcrVA5 N-terminal peptide complex to inhibit the Cas2-mediated endonuclease activity. Moreover, a detailed structural comparison between our crystal structure and homolog structure shows that binding of AcrVA5 to Cas2 causes steric hindrance to the neighboring protospacer resulting in the partial disassembly of the Cas1-Cas2 and protospacer complex, as demonstrated by electrophoretic mobility shift assay. Our study focuses on this mechanism of spacer acquisition inhibition and provides insights into the biology of CRISPR-Cas systems.

The clusters of regularly interspaced short palindromic repeats (CRISPRs) and the associated proteins serve as prokaryote adaptive immune systems to protect against invading foreign genetic elements[1–4]. A functional CRISPR-Cas system has two distinguishable components; the CRISPR locus and the Cas genes[1,5]. The CRISPR locus located on genome (either chromosome or plasmid) contains repeat sequences separated by hypervariable spacers acquired from virus or plasmid DNA. There are three steps involved in the CRISPR-Cas system for prokaryotes; new spacer acquisition into CRISPR array, crRNA transcription and effector complex biogenesis, and interference to degrade invading nucleic acids[4]. During the spacer acquisition step, the heterohexameric complex of four Cas1 and two Cas2 proteins preferentially insert a protospacer at the first CRISPR repeat region following the leader sequence[6–9]. The protospacer-adjacent motif (PAM) complementary sequence in the 3'- overhang of the protospacer is recognized by the catalytic subunits (Cas1) in a sequence-specific manner for the *E. coli* Cas1-Cas2 complex[10]. *E. fae* Cas1-Cas2 from type II-A CRISPR-Cas systems bind with protospacers with 3'-overhangs. This integration complex catalyzes a connection between the 3'-overhang of the protospacer and one strand of the repeat at leader-proximal region[7]. After half-integration intermediate occurs, nucleophilic attacks at the other site of the opposing strands on the leader-repeat border will result in single-stranded DNA gaps. These gaps are repaired by unknown mechanisms, after which full-site integration is achieved[7,8,10]. Spacer precursors trigger formation of the Cas4-Cas1-Cas2 complex, in which Cas4 helps select the PAM-flanking sequences with 3'-overhang for spacer biogenesis in directional spacer acquisition[6]. Moreover, the additional DnaQ exonuclease domain, which is fused to Cas2, significantly promotes integration[11]. The adaptation in type III-A is enhanced by the nuclease activity of AddAB (the DNA repair machinery of the cell)[12]. In the crRNA transcription stage, the transcript pre-crRNA molecule was sliced into mature crRNAs, which then bound to RNA-guide Cas proteins. In the interference stage, the mature crRNA in the surveillance complex recognizes and guides to cleave the cognate DNA or RNA.

[1]College of Veterinary Medicine, Jilin University, 130062 Changchun, Jilin, China. [2]State Key Laboratory for Diagnosis and Treatment of Severe Zoonotic Infectious Diseases, Key Laboratory for Zoonosis Research of the Ministry of Education, Institute of Zoonosis, Jilin University, 130062 Changchun, Jilin, China. ✉e-mail: mox@jlu.edu.cn

In response to this adaptive immune system, phages or prophages encode anti-CRISPR proteins (Acrs) that inhibit the adaption and/or CRISPR interference in CRISPR-Cas systems[13–17]. In literature, many Acrs have been identified, most of which interact with the Cas proteins to block their activity[18–20]. In type I CRISPR-Cas systems, primed acquisition was inhibited in the presence of AcrIF1-5[21]. Meanwhile, AcrIF1, AcrIF2, and AcrIF4 prevent the surveillance complex from interacting with target DNA[22,23]. AcrIF3 and AcrIE1 disable surveillance complex mediated recruitment of Cas3, therefore preventing it from cleaving the DNA[14,17,24]. AcrIF5 destabilizes the helical bundle domain of Cas8f in the Csy-dsDNA complex, preventing subsequent Cas2/3 recruitment[25]. In type II CRISPR-Cas systems, AcrIIA6 allosterically induce the dimerization of st1-Cas9 to reduce the binding affinity of Cas9 for DNA[26], nevertheless, truncated AcrIIA6 does not block the interference activity of Cas9 but prevents acquisition of new spacers[27]. AcrIIA2, AcrIIA4, AcrIIC3, AcrIIC4, and AcrIIC5 inhibit the Cas9-crRNA-tracRNA complex from recognizing target DNA[28,29]. AcrIIA5 and AcrIIC1 are broad-spectrum inhibitors of diverse Cas9 orthologs as they prevent DNA target cleavage[30,31]. In type V CRISPR-Cas system, AcrVA1, AcrVA4, and AcrVA5 prevent DNA recognition of Cas12-crRNA complex[32,33]. Although the capability of some Acrs to inhibit adaptation has been exploited, however, the molecular mechanism of Acr-mediated adaptation inhibition remains largely unknown.

In this study, we report and characterize an Acr which inhibits the integration in the microbial adaptive immune system. We demonstrate that AcrVA5 prevent integration of protospacer into CRISPR array in pCRISPR by gel migration assay and PCR-based assay. Through analysis and comparison of the integration inhibition by AcrVA5, we speculate that AcrVA5 may intercept the spacer acquisition via interaction with subunits in the integration complex other than Cas1, owing to that Cas1 alone catalyzes a similar low-level of half-site integration of the protospacer as that of the AcrVA5-mediated inhibition integration. To validate our assumption and examine the interaction between Cas2 and AcrVA5, we performed biochemical assays in which the AcrVA5 acetylate Cas2 at Lys[55] with direct binding to regulate the endonuclease activity of Cas2. Moreover, the crystal structure of Cas2 complex with an N-terminal peptide of AcrVA5 reveals the structural details of this interaction, in which the lysine-acid (Lys[55]) in the ferredoxin-folded Cas2 is in direct contact with the peptide ($^3$IELSG[7]) of AcrVA5. A structural comparison between our structure and the homologous Cas1-Cas2-protospacer complex reveals that AcrVA5 causes steric hindrance to the formation of the integration complex, resulting in the partial disassembly of the integration complex and failed integration.

## Results

### Identification of Acrs candidates to inhibit protospacer integration

To reconstitute in vitro protospacer integration, we adopted previous protocols with modification[8,34]. The assays were performed using purified Cas1-Cas2 complex, 30 bp protospacer DNA, and a pCRISPR plasmid, which is derived from pUC19 backbone with an inserted CRISPR locus containing leader DNA, eight repeats, and eight spacers. The integration complex (Cas1-Cas2-protospacer) catalyzes the supercoiled plasmid into three products: relaxed plasmid, linear plasmid, and Band X. In our study, the gel mobility of the migrating relaxed plasmid products was observed with the increasing protospacer concentrations, indicating Cas1-Cas2 recognized and integrated the protospacers into the pCRISPR plasmid in the integration assay (Fig. 1a). The relaxed plasmid products are composed of half-site integration (pCRISPR attack by one end of protospacer) and/or full-site integration products (pCRISPR attack by both strands of protospacer).

Next, to identify inhibitors of the type II-A CRISPR-Cas system, we expressed and purified 136 Acrs from phage or prophage genome gathered in Anti-CRISPRdb (http://guolab.whu.edu.cn/anti-CRISPRdb/ )[35]. The integration inhibition assay was performed to assess the capability of the candidates to intercept new spacer acquisition. Notably, among these candidates, *Moraxella bovoculi*-AcrVA5 (*Mb*-AcrVA5) was able to dramatically suppress the protospacer integration after pre-incubation of the integration complex with AcrVA5 (Fig. 1b). To further investigate this issue, half-site integration was detected by PCR using a forward primer in spacer-8 and a reverse protospacer primer, in which the acquisition was preferred at the leader-proximal repeat-spacer junctions by sequencing (Fig. 1c). The acquisition reaction was weakened significantly by the presence of *Mb*-AcrVA5, as is consistent with the product plasmid migration resulting from integration of protospacer DNA into the pCRISPR plasmid (Fig. 1d). Notably, Cas1 alone was able to catalyze a similar low-level of protospacer integration (Fig. 1d). Moreover, we performed in vivo protospacer acquisition and acquisition inhibition assays. These assays demonstrated that AcrVA5 could also inhibit the integration of the protospacer by the Cas1-Cas2 complex in a live setting. As depicted in Fig. 1e, the efficacy of the spacer acquisition reaction was notably diminished in the presence of AcrVA5. These in vivo results align well with those from our in vitro assay. Remarkably. despite comprehensive experimentation, we were unable to observe the effective incorporation of a spacer from the protospacer by the type V CRISPR-Cas system. In contrast, for type I-C, the protospacer sequence was effectively integrated as a new spacer into the CRISPR array in the pCRISPR plasmid (Supplementary Fig. 1). This process, guided by the proximity of conserved motifs, was highly efficient. The incorporated spacer was oriented correctly at the leader-proximal end of the CRISPR array. In literature, Cas1 active-site mutants in the protospacer acquisition assay demonstrates the catalytic role of Cas1 during spacer acquisition[36]. Therefore, we speculate that AcrVA5 is unlikely to suppress the catalytical activity of Cas1 during adaptation in vitro.

### Direct interaction between AcrVA5 and Cas2 in vitro

The in vitro integration inhibition assay indicates that AcrVA5 is capable of inhibiting the protospacer acquisition, however, pre-incubation of AcrVA5 with integration complex did not significantly affect Cas1's integrase activity. To identify binding partners of AcrVA5, glutathione S-transferase (GST) pull-down assays were carried out using N-terminal GST-tagged AcrVA5, as well as Cas2 and Cas1 with and without protospacer bounds. As expected, neither the free Cas1 nor the protospacer-bound Cas1 was found to interact with AcrVA5 (Supplementary Fig. 2). Thus, AcrVA5 does not bind to interfere with the activity of Cas1, at least with regard to its integrase activity. However, the selective elution from GST affinity beads resulted in the co-elution of both GST-AcrVA5 and Cas2 proteins, indicating that there is an interaction between AcrVA5 and Cas2 (Fig. 2a). To further examine whether or not this interaction is direct and quantify the binding affinity, isothermal titration calorimetry (ITC) experiments were performed. The dissociation constant (Kd) of the interaction between Cas2 and AcrVA5 was determined to be 1.1 μM (Fig. 2b). These results confirm that AcrVA5 inhibits the spacer acquisition through direct interaction with Cas2 in the integration complex.

### Inhibited the endonuclease activity of Cas2 by AcrVA5

In literature, Cas1 is the catalytic subunit, whereas Cas2 merely served as a structural platform for the integration complex assembly to aid with protospacer binding and substantially promote the integration in the adaptation step[36]. Meanwhile, the endonuclease activity of the Cas2 proteins from other species was described[37–39]. *Ss*-Cas2 has been described as a single-strand endoribonuclease that requires metal ions to function and shows a preference for U-rich ssRNA[40]. *Bh*-Cas2, on the other hand, is recognized for its endonuclease activity targeting double-stranded DNA substrates[37].

In this study, Cas2 is capable of cleaving dsDNA in non-specific manner (Supplementary Fig. 3a). Moreover, the addition of EDTA completely inhibits plasmid degradation, indicating that Cas2 is a

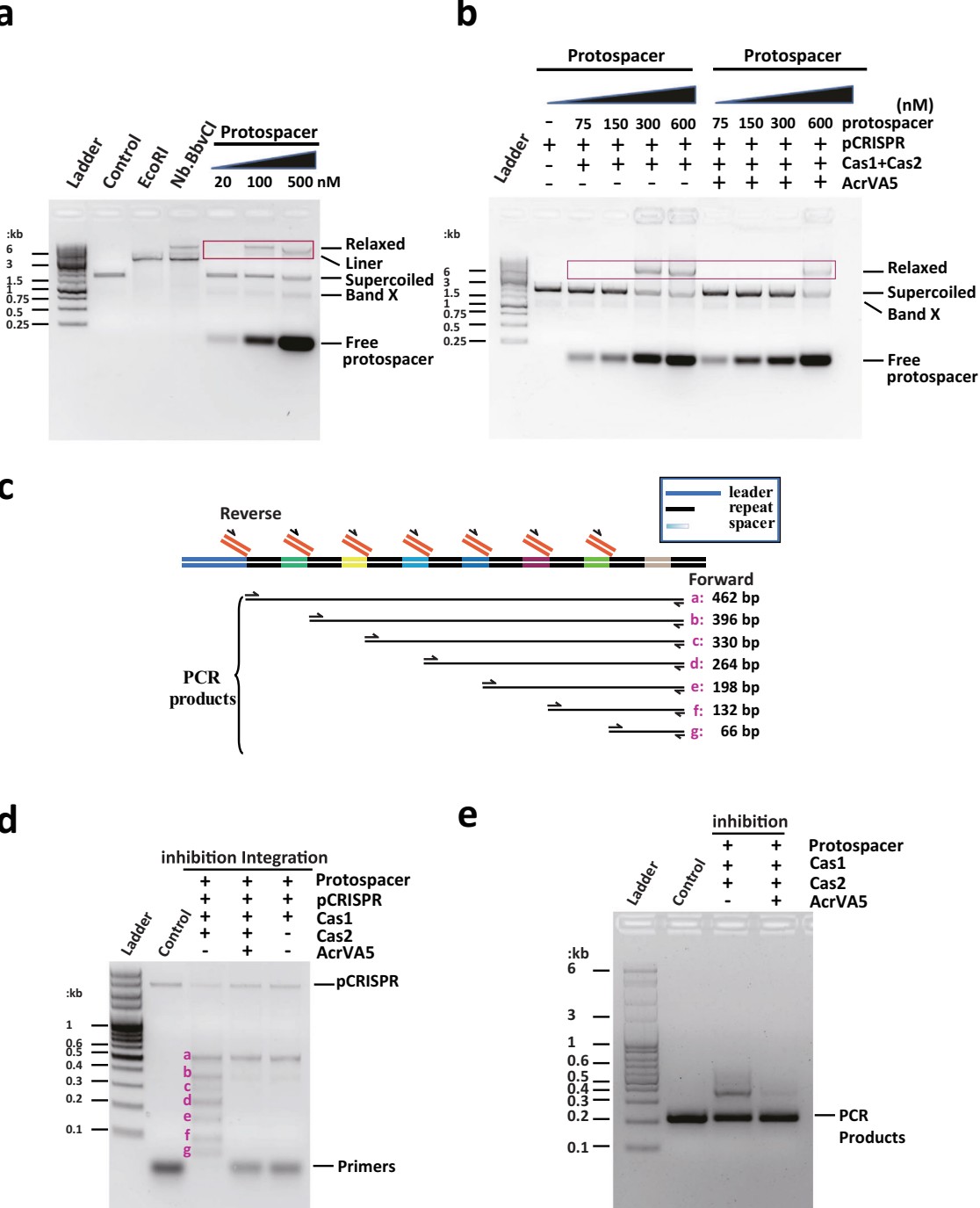

**Fig. 1 | Protospacer Integration assay and integration inhibition assay. a** Similar to the *Eco*RI- and Nb.*Bbv*CI nickase-treated pCRISPR, integration assay generated linearized and supercoiled plasmids of the integration products. The relaxed plasmids product migrated with increasing protospacer concentrations. The integration experiments have been repeated independently for three times with similar results. **b** The DNA product migration is labeled in red box. Left panel: the gel mobility of the slowly migrating DNA product catalyzed by Cas1-Cas2 was observed. Right panel: By the presence of AcrVA5, integration reaction weakened significantly, indicating AcrVA5 inhibit the integration activity of the Cas1-Cas2. The integration inhibition experiments have been repeated independently for three times. **c** Schematic of the half-integration assay. Cas1-Cas2 catalyzed protospacer

integration into the CRISPR array in the pCRISPR plasmid. The half-site integration was detected by PCR and the size of the predicted PCR products were shown. **d** A 30 bp protospacer was incubated with Cas1-Cas2 and pCRISPR for 60 min and integration was detected by PCR and gel electrophoresis (Lane 2). Lane 3 showed the acquisition inhibition assay by AcrVA5. The PCR fragments difference indicates the inhibited half-integration of protospacer into pCRISPR plasmid. Lane 4 showed the low-level of integration catalyzed by Cas1 protein only. **e** This figure visually depicts the inhibitory effect on protospacer integration within a live environment (in vivo). Both the in vitro and in vivo integration experiments have been repeated independently for three times with similar results. Source data are provided as a Source Data file.

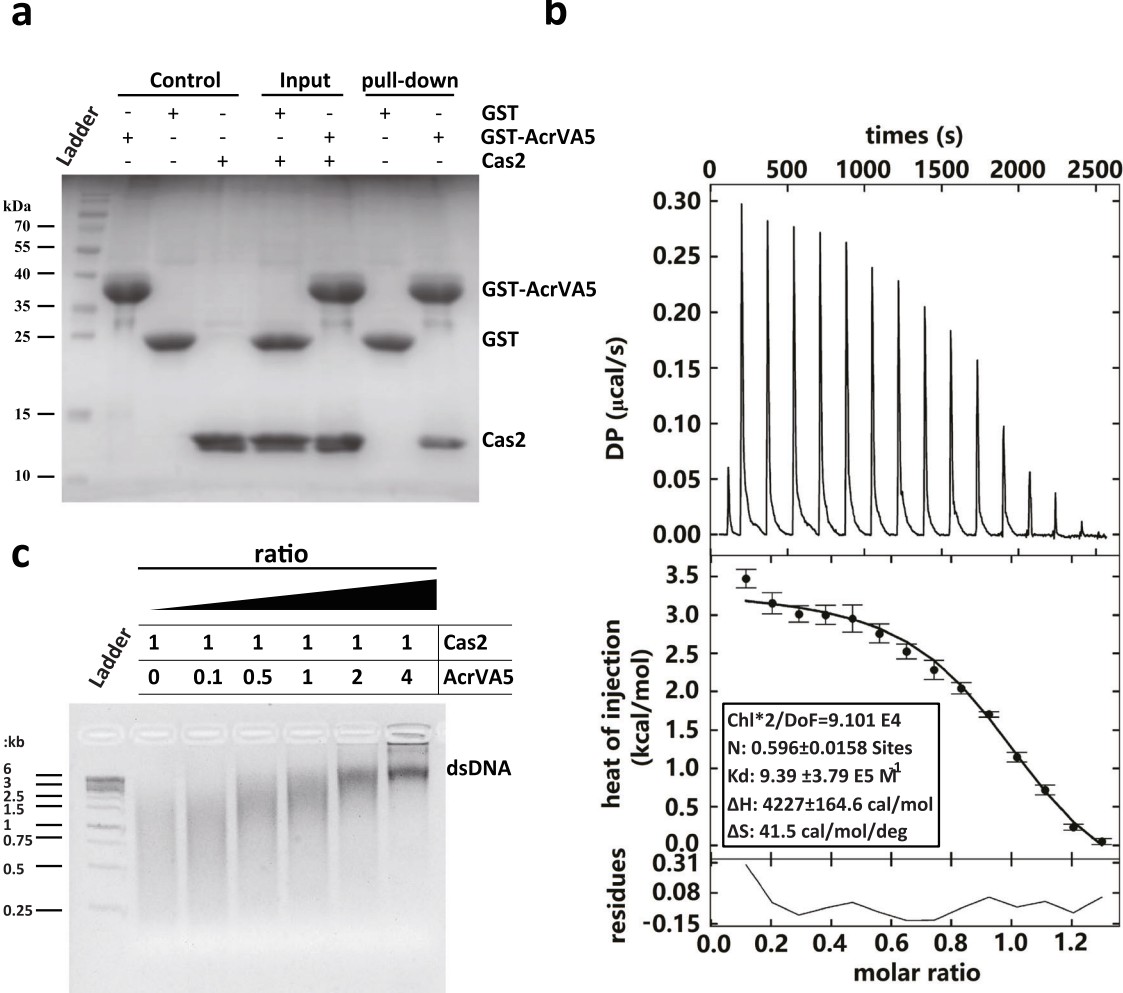

**Fig. 2 | Interaction between Cas2 and AcrVA5. a** Direct interaction between Cas2 and AcrVA5 in the in-vitro GST pull down assay. The GST-AcrVA5 bound Glutathione Agarose resin was incubated with Cas2 protein. After excessive washing, the protein resolved was analyzed in SDS-PAGE and detected by Coomassie brilliant blue staining. The pull-down experiments reported in this study have been repeated for three times. **b** Isothermal Titration Calorimetry of Cas2 with AcrVA5. The ITC trace of AcrVA5 injection into a Cas2-containing sample cell. All of the ITC experiments reported in this study have been repeated for three times. $N = 0.596 \pm 0.0158$, Kd (dissociation constant) $= 9.39 \pm 3.79$ E5 M$^{-1}$, ΔH (Δ Enthalpy) $= 4227 \pm 164.6$ cal/mol and ΔS (Δ Entropy) $= 41.5$ cal/mol/deg. The $N$ and $K$ values calculated in one of three independent experiments are reported. **c** Inhibition of Cas2 nuclease activity by AcrVA5. Gradient course of the dsDNA cleavage assay was inhibited by AcrVA5, in which the representative ratio is shown at the top of each lane. The inhibition experiments have been repeated for three times with similar results. Source data are provided as a Source Data file.

metal-dependent ribonuclease. To identify Cas2's preferred divalent ions, an ion dependency survey in the endonuclease activity of Cas2 was performed. Results indicate that $Mn^{2+}$ ions were more preferred than $Mg^{2+}$ ions, which in turn were more preferred than other divalent ions (Supplementary Fig. 3b). Next, the cleavage assay against dsDNA was performed to determine the preferred pH for dsDNAse activity and is identified to be between pH 7.0 and 10.0 (Supplementary Fig. 3c). To explore the AcrVA5-mediated cleavage inhibition of Cas2, the endonuclease activity of Cas2 was tested in the presence of AcrVA5. Notably, enhanced inhibition of the plasmid DNA cleavage was observed with increasing amount of AcrVA5. In particular, the endonuclease activity of the Cas2 protein was completely abrogated by the pre-incubation of Cas2 with AcrVA5 (molar ratio: 1:4) for 30 min (Fig. 2c). Hence, the data suggests, rather unexpectedly, that AcrVA5 has weak binding affinities towards Cas2 and can substantially inhibit the dsDNAse activity of Cas2. In addition, we examined the inhibitory effect of *Mb*-AcrVA5 on several proteins: *Mb*-Cas2 from the type I-C CRISPR-Cas system and *Mb*-Cas2 from type V-A CRISPR-Cas system. Notably, AcrVA5 was able to inhibit the dsDNAse activity of *Mb*-Cas2.

Although the endonuclease activity of the *Mb*-Cas2 proteins from the type V-A CRISPR-Cas system was observed, *Mb*-AcrVA5 acted as a broad-spectrum inhibitor for Cas2 across various species, including *Treponema denticola* and *Moraxella bovoculi* (Supplementary Fig. 4).

**The crystal structure of Cas2 and AcrVA5 peptide complex**
We endeavored to uncover the interaction between proteins Cas2 and AcrVA5 through a structural perspective by attempting a co-crystallization of AcrVA5 and Cas2. Despite rigorous crystallization trials, we were unable to obtain suitable crystals of the Cas2-AcrVA5 complex. Concurrently, we synthesized six peptides, each corresponding to different but equal parts of the AcrVA5 amino acid sequence. We then analyzed the binding affinity of each peptide to Cas2 using a peptide scanning affinity capture assay. As expected, a stable complex was formed when Cas2 was paired with the first peptide ([1]MKIELSGGYICYSIE[16]) (Supplementary Fig. 5a–d). No other stable Cas2-peptide complexes were detected, suggesting a specific interaction between Cas2 and this segment of AcrVA5. Moreover, we measured the heat absorbed or released during the interaction between

the AcrVA5-peptides (peptide-2-peptide-6) and Cas2. It appeared that either the binding affinity is exceptionally weak, or there is no interaction occurring between Cas2 and the others AcrVA5-peptide (Supplementary Fig. 6A–E).

To further validate the functional importance of the N-terminal region of AcrVA5 for Cas2 binding, we have conducted ITC to study the binding of Cas2 to the L5A point mutant and the M1-G7 deletion mutant of AcrVA5. In the ITC experiment, we measured the heat absorbed or released as Cas2 interacts with both the AcrVA5-L5A and M1-G7 deletion mutant of AcrVA5. Our findings revealed that the dissociation constant (Kd) of the interaction between Cas2 and AcrVA5-L5A was 1.3 μM (Supplementary Fig. 6a–f). However, integrating the individual peaks for the M1-G7 deletion mutant of AcrVA5 with the software proved to be a challenge (Supplementary Fig. 6g). In addition, we conducted both in vitro and in vivo integration assays to confirm the functional significance of the N-terminal region of AcrVA5 for Cas2 binding. During these procedures, adding AcrVA5-L5A to the Cas1-Cas2-protospacer led to less amount of the protospacer being incorporated into the plasmid compared to the untreated Cas1-Cas2-protospacer (Supplementary Fig. 7a, b). Conversely, the amount of protospacer incorporated into the pCRISPR plasmid by mixing the M1-G7 deletion mutant of AcrVA5 with the Cas1-Cas2-protospacer complex resembled that of the untreated Cas1-Cas2-protospacer complex (Supplementary Fig. 7a, b). Hence, the Cas2-peptide ([1]MKIELSGGYICYSIE[16]) complex was chosen for structural determination. The complex of Cas2 with the N-terminus peptide candidates of AcrVA5 was crystallized in the space group of $P2_12_12_1$, with following unit-cell parameters: $a = 26.592$ Å, $b = 76.391$ Å, $c = 88.445$ Å and $\beta = 90°$. The structure of the Cas2-peptide ([3]IELSG[7]) complex was determined by Molecular Replacement at 2.0 Å resolution, using 5XVN as the search model[41]. The initial model was further refined by Phenix/refine and rebuilt using Coot, with the final $R_{factor}$ and $R_{free}$ values of 19.51% and 23.21%, respectively. The complex structure contained one Cas2 dimmer and one AcrVA5 peptide per asymmetric unit, with crystallographic statistics listed in Table 1. The stoichiometry of Cas2 relative to AcrVA5, as derived from our ITC experiments, was approximately 0.596. These findings align with the structural observations from our high-resolution (2.0 Å) structure of the complex formed by a Cas2 dimer with a AcrVA5 peptide, highlighting the consistent one-to-two ratio between AcrVA5-peptide and Cas2 dimer.

The crystal structure of Cas2 reveals a compact dimer structure (protomer A & B) in which each chain is composed of three α-helices and five β-strands (α1: residues 14–29, α2: residues 47–59, α3: residues 74–78, β1: residues 1–8, β2: residues 33–36, β3: residues 39–44, β4: residues 66–73 and β5: residues 81–84) (Fig. 3a, left panel). For AcrVA5-peptide, we have produced an omit map visualized in Supplementary Fig. 8. In this respective structure, the model of the AcrVA5-peptide (with discernible and constructed residues: Ile-Glu-Leu-Ser-Gly) is represented as a ball and stick model. In our structure, the interaction between the AcrVA5-peptide and Cas2 is facilitated by a series of interactions: a hydrogen bond between Lys[59] of Cas2 and Ile[3] of AcrVA5, as well as non-bond contacts between Lys[55] of Cas2 and Leu[5] of AcrVA5, Gln[58] of Cas2 and Leu[5] of AcrVA5, and Lys[59] of Cas2 and Ile[3] of AcrVA5. The distances between these interacting atoms and a 2D representation of these binding interactions are detailed in Fig. 3a, right pane, Supplementary Fig. 9a, b. The buried surface area of the complex of Cas2 and the AcrVA5 peptide is 434.56 Å² per molecule (Supplementary Table 1), thereby suggesting that this complex could be physiologically relevant. Structure superimposition between two protomers of Cas2 reveals a high similarity in the overall structure with a ferredoxin-like fold with an r.m.s.d of 2.03 Å (overall 88 residues), except for the significant structural conformation variation in the C-terminus (Supplementary Fig. 10). The structural heterogeneity observed in the C-terminus was most likely caused by the interaction between Cas2 and the peptide of AcrVA5. In literature, the C-terminus

**Table 1 | Data collection and refinement statistics of structure of the complex of Cas2 and peptide of AcrVA5**

| Data collection | Cas2 in complex with AcrVA5 |
|---|---|
| Space group | $P2_12_12_1$ |
| PDBID | 8IA4 |
| Wavelength (Å) | 1.000 |
| Cell dimensions | |
| a (Å) | 26.592 |
| b (Å) | 76.391 |
| c (Å) | 88.445 |
| α, β, γ (°) | 90, 90, 90 |
| Molecule/ASU | one Cas2 dimer and an AcrVA5-peptide |
| Resolution range (Å)[a] | 19.75–2.0 (2.07–2.0) |
| $R_{sym}$ (%)[a] | 7.65 (35) |
| I/(I) | 13.5 (3.5) |
| Completeness (%)[a] | 99.38 (96.21) |
| Redundancy[a] | 10.5 (9) |
| *Refinement* | |
| Search Model | 5XVN |
| Resolution (Å)[a] | 2.0 (2.072–2.0) |
| No. reflections | 12,755 (1217) |
| $R_{work}$ ($R_{free}$) (%) | 19.51/23.21 (18.64/24.90) |
| No. atoms | 1766 |
| Protein | 1568 |
| Water | 198 |
| B-factors (Å²) | 18.69 |
| Protein | 19.08 |
| Water | 28.3 |
| R.m.s. deviations | |
| Bond lengths (Å) | 0.006 |
| Bond angles (°) | 0.99 |
| % favored (allowed) in Ramachandran plot | 96.24 (3.76) |

[a]Values for the highest-resolution shell are in parentheses.

of Cas2 is critical for the function of the spacer acquisition because deletion of the C-terminus removes the backbone interaction of Cas2 with the β4 strand of Cas1, thereby causing disassembly of the Cas1-Cas2-dsDNA hexamer complex[36].

To elucidate the structural mechanisms of the AcrVA5-mediated inhibition of integration, we superimposed our crystal structure with the pre-existing structures of the Cas1-Cas2-protospacer complex (PDB: 5XVN) and AcrVA5 (PDB 6IUF), which was achieved by aligning the AcrVA5-peptide and AcrVA5. In this pseudo full complex structure, the central part of the dual-forked protospacer intervene the concave region between two α1 of the dimeric Cas2, and the phosphate backbone of protospacer interact with the positively charged residues in the α1 of Cas2, with α1 reaching in the major grove of the protospacer. In our structure, the peptide of AcrVA5 bind with the lysine-acids in the α2, which is adjacent to the α1 in Cas2. Although the β6–β7 region in the structure of Cas2 is unresolved, the binding of AcrVA5 in the α2 helix of Cas2 causes steric hindrance with the neighboring protospacer (Fig. 3b). To further confirm that the decrease in integration efficiency was indeed caused by AcrVA5, the protospacer binding activity of Cas1-Cas2 complex was tested in the presence of AcrVA5 via bio-dot electrophoretic mobility shift assays (EMSA). Interestingly, despite the fact that interaction between the protospacer and Cas1-Cas2 induces conformational bending of the DNA[10], co-incubation of AcrVA5 with integration complex weakened the binding affinity between Cas1-Cas2

**Fig. 3 | Crystal structure of the Cas2 and the peptide of AcrVA5. a** Cartoon representation of crystal structure of complex of Cas2 and AcrVA5-peptide. The crystal structure of Cas2 reveals a compact dimer structure (protomer A & B), in which the N-terminal peptide of AcrVA5 bind to the α2 helix of protomer B. The protomer A and protomer B are labeled in red and cyan respectively. The bound peptide of AcrVA5 is represented in stick mode (left panel). Close-up view of molecular interactions between Cas2 and the peptide of AcrVA5. The side chains of the conserved interaction residues are indicated (right panel). **b** Cartoon representation of visualization structural model of Cas1-Cas2 in complex with peptide of AcrVA5 derived from structural superimposition of structures of Cas2-AcrVA5 peptide with Cas1-Cas2 in complex with protospacer. The bound peptide of AcrVA5 is represented in shaded area, which may interfere interaction of Cas1-Cas2 with protospacer and cause an allosteric inhibition of the prime acquisition activity of the integration complex. **c** Bio-dot EMSA to test the inhibition effect of AcrVA5 on protospacer-binding activity of Cas1-Cas2, in which the interaction of AcrVA5 and Cas2 lead to the partial disassembly of the integration complex. The bio-dot EMSA have been repeated for three times with similar results. Source data are provided as a Source Data file.

and the protospacer (Fig. 3c). The binding of AcrVA5 to Cas2 generates steric hindrance with the neighboring protospacer, thereby resulting in the partial disassociation of the protospacer, and subsequently, the disassembly of the integration complex as demonstrated by bio-dot EMSA (Fig. 3c).

## The suppression of Cas2 dsDNAse activity by AcrVA5, likely through acetylation of residue-Lys[55]

In the dsDNAse inhibition assay, the addition of AcrVA5 dramatically reduced the cleavage activity of Cas2. We speculate that the endonuclease activity difference is attributed to the AcrVA5-mediated modification of Cas2, owing to the fact that AcrVA5 functions as a broad-spectrum acetyltransferase and influences a large number of cellular metabolic processes[42]. To investigate this issue, AcrVA5 and Cas2 were co-expressed in *E. coli,* and the Cas2 protein was further purified for mass spectrometry (MS) analysis. The MS results of the Cas2 proteins before and after being treated by AcrVA5 show that the lysine-acid (Lys[55]) is most likely the acetylation sites of AcrVA5 in vivo (Fig. 4a and Supplementary Fig. 11). Based on the structural information from our crystal structure and the electrostatics surface analysis, mutations were introduced to critical residues (E15, K42, L45, K55, S56, K59 and E64) (Fig. 3a and Supplementary Fig. 12). To identify the potential catalyzed residues in the Cas2 protein, cleavage assay was performed by comparing the endonuclease activity among the wild type (WT) and

mutants. Interestingly, the introduction of the Alanine mutation at conserved L45 (located in the loop region between α2 and β3) disrupted the cleavage activity of Cas2. Other mutants (E15Q, K42A, and E64A) displayed compromised endonuclease activity against dsDNA (Fig. 4b, Supplementary Fig. 13). To further examine the impact of the potential catalytic residues in Cas2 protein by AcrVA5, cleavage inhibition assay was performed to test whether the residues of Cas2 in the binding interface play a role in determining the mechanism of AcrVA5 inhibition. Consistent with the MS results, the K55A mutant rendered Cas2 insensitive to inhibition by AcrVA5 in degradation dsDNA substrates, which shows that Lys[55] is a structural determinant dictating Cas2 inhibition by AcrVA5 (Fig. 4c). Despite the possibility that the inhibition of Cas2's endonuclease activity could be attributed to the acetylation of a lysine-acid (Lys[55]) by AcrVA5, it is noted that these lysine acids are partially conserved in other homologs in the CRISPR-Cas system (Supplementary Fig. 13). Our data suggests that AcrVA5 may affect the endonuclease activity of Cas2 in CRISPR-Cas systems through the catalyzed acetylation of the lysine-acid (Lys[55]).

## Discussion

Bacteria and archaea rely on CRISPR-Cas systems to defend against harmful foreign genetic materials, includes plasmids and bacteriophages et al. The evolutionary 'arms race' between bacteria and phages provides a selective pressure for the emergence of inhibitors of

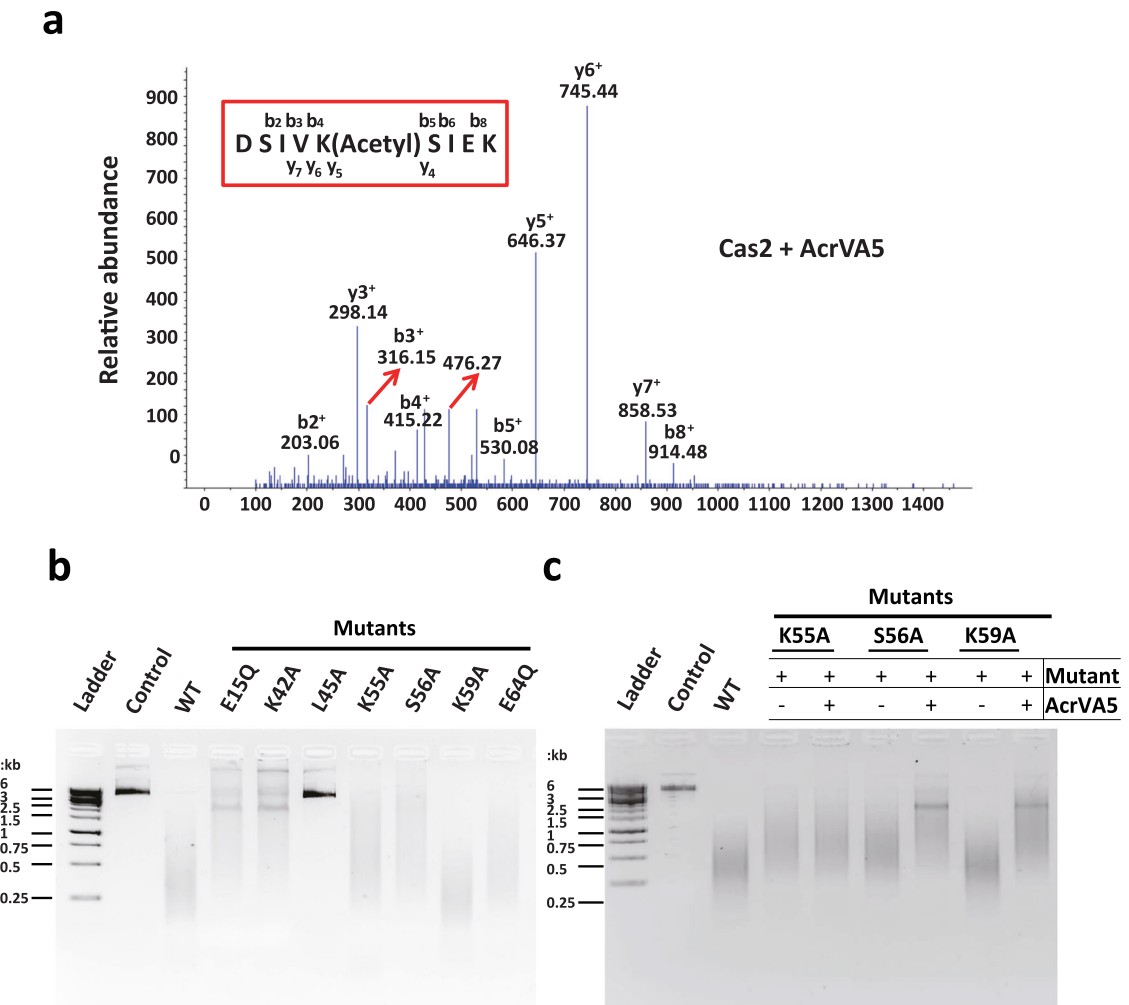

**Fig. 4 | The acetylation modification identified in mass spectrum of Cas2 co-expressed with AcrVA5 in *E. coli*. a** The mass spectrometry of Cas2 co-expressed with AcrVA5 in *E. coli*. The mass spectrum charged ion showed that Lys[55] is acetylated in the peptide DSIVKSIEK. The b and y ions indicate the peptide fragmentations containing the N terminus and C terminus of the peptide, respectively. **b** Agarose gel of in vitro dsDNA cleavage assays. The WT and mutations of Cas2 at the continuous positive patch in the electrostatic interface as indicated by labeling

of these residues in Supplementary Fig. 10. The in vitro dsDNA cleavage assays have been repeated for three times. **c** Cleavage inhibition assays used to test the inhibition activities of AcrVA5 against WT and the mutants (K55A, S56A and K59A) of Cas2. The dsDNAse activity of K55A mutant is insensitive to the inhibition of AcrVA5. The cleavage inhibition assays have been repeated for three times. Source data are provided as a Source Data file.

CRISPR-Cas systems, known as Acrs. Many Acrs were found in virulent phages, temperate phages, prophage remnants, and horizontally acquired genomic islands[43]. If activated, the prophage cuts itself out of the bacterial genome, replicates, and then packages its DNA into phages which could go on to infect other bacteria[44,45]. Prophages are major contributors to horizontal gene transfer and drive the evolution and diversification of bacteria[46]. CRISPR-Cas systems are sometimes also present in prophages regions. Bacteriophages can transduce CRISPR-Cas systems between bacteria, which can offer immunity against other phages and thereby endow them with an advantage over competing phages[46,47]. Therefore, bacteriophages may use an extensive battery of counter-defense strategies to co-exist in the presence of CRISPR-Cas defense mechanisms in different bacteria strains.

In literature, AcrVA5 is a type V anti-CRISPR protein, located in the prophage regions of *Moraxella bovoculi* strains and serves as an N-acetyltransferase[35]. There are a large number of cellular proteins, including Cas12a in type V CRISPR-Cas system, which could be acetylated by AcrVA5[33,42]. In this study, we identified two new functional roles of AcrVA5: (1) *Mb*-AcrVA5 can inhibit the protospacer integration by the Cas1-Cas2 complex, primarily via binding and causing steric hindrance. (2) *Mb*-AcrVA5 could suppress the dsDNAse activity of both

*Mb*-Cas2 from type I-C and *Td*-Cas2 from type II-A though acetylation of Lys[55]. Our biochemical and structural data indicate that AcrVA5 primarily inhibits integration by the Cas1-Cas2 complex via binding and ensuing steric hindrance. To test the protospacer binding capacity to the Cas1-Cas2 complex, we performed bio-dot EMSA both in the absence and presence of AcrVA5. It was observed that when AcrVA5 was co-incubated with the integration complex, the binding affinity between the Cas1-Cas2 complex and the protospacer was compromised (Fig. 3c). Furthermore, the binding of AcrVA5 to the α2 helix of Cas2 creates steric hindrance with the adjacent protospacer (Fig. 3b). Additionally, we conducted an integration assay to evaluate the impact of Lys[55] acetylation on the in vitro integration activity of Cas1-Cas2. It was observed that the Cas1-Cas2-protospacer complex, post-acetylation mediated by AcrVA5, demonstrated an integration capacity akin to the untreated Cas1-Cas2-protospacer complex, incorporating a similar amount of the protospacer into the plasmid (Supplementary Fig. 14). As a result, acetylation modifications on Cas2 seem to produce a negligible effect on protospacer integration. The MS results revealed that upon Cas2 protein treatment with AcrVA5, the amino acid lysine at position 55 (Lys[55]) is likely the in vivo acetylation site of AcrVA5. In addition, creation of a K55A mutant

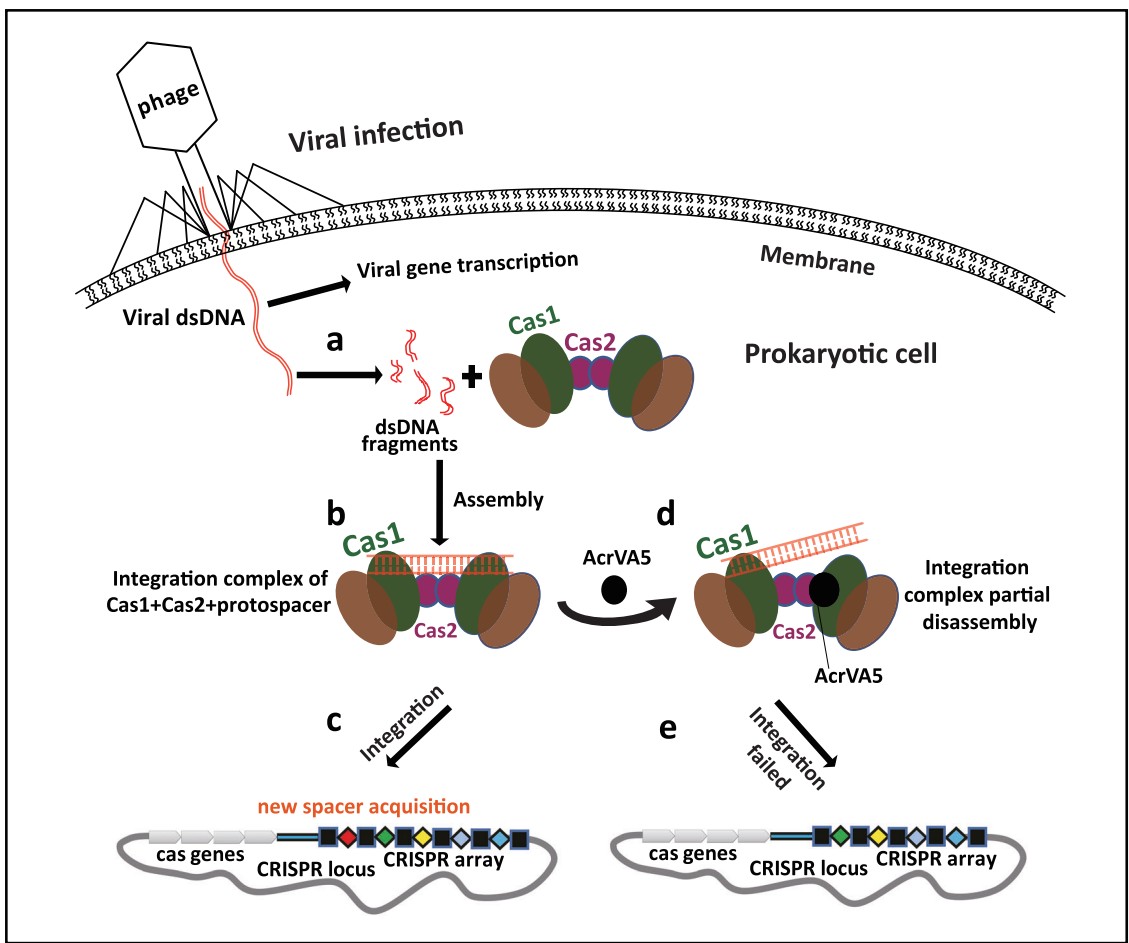

**Fig. 5 | The model for roles of Cas2 in spacer acquisition during CRISPR-Cas adaptive immunity. a** Structural model of Cas1-Cas2 complex at free state and the process of productive protospacer biogenesis. The structure models of Cas1 are colored in brown and green, respectively. The structure model of Cas2 is colored in purple, and the dsDNA fragments are colored in red. **b** Structural model of integration complex at "functional spacer integration" state. **c** Integration complex catalyzed new spacer acquisition preferentially at the leader end in CRISPR array. **d** The model for partial disassembled integration complex. AcrVA5 bind to Cas2, in which the interaction generates steric hindrance with neighboring protospacer. **e** Partial disassembled integration complex failed to catalyze new spacer acquisition in CRISPR array.

resulted in Cas2 becoming resistant to inhibition by AcrVA5 when degrading dsDNA substrates. This suggests that the residue Lys[55] plays a pivotal role in mediating the suppression of Cas2 by AcrVA5 (Fig. 4c). From these combined data, we propose that the suppression of Cas2's dsDNAse activity by AcrVA5 may predominantly occur via acetylation at the Lys[55] residue. Although AcrVA5 was known to function as an acetyltransferase, modifying a lysine residue in Cas12a during the interference step in the type V CRISPR-Cas system, AcrVA5 was also found to inhibit both the dsDNAse activity of *Mb*-Cas2 from type I-C and *Td*-Cas2 from type II-A (Fig. 2c and Supplementary Fig. 4), as well as the integration process for both types, as reflected in Fig. 1d and Supplementary Fig. 1. Based on these observations, we posited that AcrVA5 inhibits the dsDNA-cleavage activity of both Cas12a (V-a) and Cas2 (II-A) using a similar mechanism, i.e., by acetylation of a lysine residue in the active site.

Cas1 and Cas2 nucleases are the only Cas proteins found in all CRISPR-Cas systems. In type-II CRISPR-Cas system, two dimeric Cas1 protein, a dimeric Cas2 protein, and a protospacer assemble into a hexamer-dsDNA complex to facilitate the foreign genetic material integration into the CRISPR locus region during spacer acquisition[8,36]. Within the Cas1-Cas2 complex, the main function of Cas2 is usually assumed to be for the formation of a non-catalytic scaffold within the integration complex[36]. However, Cas1's ability to assemble with the Cas2 protein is also essential for spacer acquisition, because the

*Sulfolobus solfataricus* integration complex with Cas2 mutant (R18A) has low spacer acquisition levels, in which the Arg[18] is located at the inter-protein interface with Cas1[40]. Considering the C-terminus conformational changes of Cas2 in the integration complex and the conserved residues in the Cas1-Cas2 interface required for complex stability and integration, it is possible that the importance of the role of Cas2 as a central structural component in the integration complex is underestimated. In this study, our findings show that AcrVA5 affects the protospacer acquisition by disassembling the integration complex (Fig. 3c), in which AcrVA5 directly interacts with Cas2 and causes steric hindrance with the neighboring protospacer (Figs. 2a and 3b). The dissociation constant (Kd) for the interaction between Cas1 and Cas2 is reported to be ~290 nM as measured by ITC[36]. However, our study found the dissociation constant (Kd) for the interaction between Cas2 and AcrVA5 to be significantly larger, at ~1.1 μM. A smaller dissociation constant suggests a stronger binding affinity. Therefore, the binding between Cas1 and Cas2 is tighter compared to that between Cas2 and AcrVA5. To illustrate our insights into the roles of Cas2 in foreign DNA acquisition during CRISPR-Cas adaptive immunity, a visualization model was produced to show protospacer acquisition in a host cell. In this model, upon entry into the host cell, the phage genome is released from the capsid, followed by AcrVA5 being expressed and viral protein synthesis being initiated by the host cell's translational mechanisms. Meanwhile, the invading DNA is degraded by host nucleases

enzymes[10]. Together with a dsDNA fragment (protospacer) from the phage, a Cas2 dimer recruits two dimeric Cas1 proteins to the leader sequence region through an indirect mechanism[36], thereby assembling into a hexamer-dsDNA complex and for spacer acquisition. AcrVA5 directly interacts with and acetylates Cas2 proteins at Lys[55] to inhibit dsDNA cleavage activity and dissemble the Cas1-Cas2-dsDNA complex, thus, resulting in failed integration. Based on this model, we propose that phages may use Acr to suppress adaptation in CRISPR-Cas systems and prevent acquisition of new invader-derived spacers into the CRISPR array in the host genome during the infection, consequently facilitating phage survival against CRISPR-Cas attacks (Fig. 5a–e).

Notably, our in vitro protospacer integration could be detected as early as 10 min and optimally in ~1 h, however, the largest of the CRISPR-Cas systems contribute to only about 1% of the total size of the Bactria genome (*Methanocaldococcus sp.* FS406-22 and *Sulfolobus tokodaii* str. 7)[48]. Over the course of a couple of a million years, most prokaryotes maintain a few to hundreds of spacers in CRISPR-Cas systems in Bacteria-phages coevolution. Moreover, successful acquisition of new spacers against phages is a rare event, ~1 in $10^7$ cells[49]. It seems that there are dynamic determinants for the optimal number of spacers in CRISPR-Cas systems. The commonly accepted explanations are that CRISPR-Cas systems have evolved to avoid toxic levels of autoimmunity by limiting the rate of spacer acquisition in vivo, or that the spacer acquisition is regulated in prokaryotes to balance the benefits of adaptive immune protection of CRISPR-Cas systems with autoimmunity. In our study, AcrVA5 can prevent protospacer integration by Cas1-Cas2 in the microbial adaptive immune system that is likely to be shared across all types of CRISPR-Cas systems. Thus, Acrs encoded by phages or prophages may play an important role in the regulatory process for acquisition of the optimal number of spacers.

## Methods

### Cloning, protein expression and purification of Cas2, Cas1 and AcrVA5 protein

We amplified the *Treponema denticola* Cas2 gene using Phusion DNA polymerase (Thermo Fisher Scientific) and subsequently cloned it into a modified pET32a vector. This vector was specifically adapted to include a precession enzyme cleavage site, enhancing the expression of an N-terminal Trx-tagged, soluble protein. The Cas1 and AcrVA5 genes were also amplified and cloned into a pET28b vector separately, each featuring a C-terminal His-tag. Similarly, GST-tagged Cas2 and GST-tagged AcrVA5 genes were cloned from corresponding DNA sequences into the pGEX-6p-1 vector, each possessing an N-terminal GST-tag. For MS analysis, Cas2 and AcrVA5 genes were individually sub-cloned into a pET-Duet vector, with AcrVA5 equipped with a C-terminal His-tag. A pCRISPR plasmid was constructed by amplifying the CRISPR locus from the *Treponema denticola* genome and inserting it into the pUC19 (All the plasmids and primers used in the study are listed in the sources data file). *E. coli* BL21 (DE3) transformed with the recombinant plasmid were cultured in an LB medium supplemented with Kanamycin/Ampicillin antibiotics. Upon reaching an $OD_{600}$ of 0.6, recombinant protein expression was induced by the addition of a final concentration of 0.4 mM IPTG. After overnight culture, cells were collected by centrifugation and resuspended in a lysis buffer (20 mM Tris, pH 7.4, 500 mM NaCl, 1 mM DTT, and 2 mM EDTA). The suspension was homogenized four times while being kept on ice and subsequently subjected to ultracentrifugation at 31,237 g for 1 h. The supernatant collected was then purified for the Cas2 protein by Histidine chromatography, followed by affinity tag removal, Heparin chromatography, and size-exclusion chromatography (ClearFirst-3200). Cas1 and AcrVA5 proteins were purified using a similar method but did not require affinity tag removal. GST-tagged Cas2 and AcrVA5 proteins underwent GST affinity chromatography followed by GST-tag removal and size-exclusion chromatography for purification. Finally,

co-expressed AcrVA5 and Cas2 were co-purified by Histidine affinity chromatography and size-exclusion chromatography.

### Cas2 mutant proteins preparation and Cas1-Cas2 complex formation

Cas2 mutants (K42A, L45A, K55A, S56A, K59A, and E64Q) were created using the Quick-Change site-directed mutagenesis kit and their sequences were verified. We amplified the Cas2-E15Q gene from the Cas2 gene and sub-cloned it into a pET32a vector with an N-terminal Trx-tag. The purification process of Cas2 mutant proteins closely matched that of the WT-Cas2 preparation. The purified Cas1 and Cas2 proteins were dialyzed against a buffer composed of 150 mM KCl, 20 mM Hepes (pH 7.5), 1 mM TCEP, 5% glycerol, and 10 mM imidazole. To form the complex, the purified Cas1 and Cas2 were co-incubated at 4 °C for 30 min.

### In vitro protospacer acquisition assays, and acquisition inhibition assays

Protospacer acquisition assays were performed in vitro, as previously described, with some modifications[8]. Oligonucleotides, acting as double-stranded DNA protospacers, were dissolved in a buffer containing 20 mM Heps (pH 7.5) and 25 mM KCl, then annealed by heating at 95 °C and slowly cooling to room temperature. The sequences of the 30 bp protospacers used in this study were strand 1 and strand 2 with 3' overhangs. The linearized plasmid was generated using *Eco*RI (Thermo Fisher) digestion, and the nickase-treated pCRISPR was digested with Nb.*Bbv*CI (NEB). For the integration reaction, we employed a buffer comprising 20 mM Hepes (pH 7.5), 25 mM KCl, 10 mM MgCl$_2$, 1 mM DTT, and 10% DMSO. A complex for integration was assembled by incubating 75 nM Cas1-Cas2 proteins with 200 nM protospacer in the reaction buffer at 4 °C for 30 min. This was followed by the addition of the pCRISPR plasmid at a final concentration of 7.5 nM and incubation at 37 °C for one hour. For the protospacer acquisition inhibition assay, 300 nM of AcrVA5 was pre-incubated with 225 nM of Cas2 proteins in the reaction buffer at 4 °C for 30 min. This was followed by the addition of 450 nM Cas1 and incubation at 4 °C for an additional 30 min to complete the Cas1-Cas2 assembly. Subsequently, the protospacer DNAs were incubated with the incumbent protein(s) for 30 min for integration assembly, followed by incubation with the pCRISPR plasmid mixture at 37 °C for one hour. The reaction process was stopped using 0.4% SDS and 25 mM EDTA, following which the pCRISPR plasmid was extracted with phenol-chloroform and run on a 1.2% agarose gel. The isolated pCRISPR plasmid was then used as a template for PCR amplification. It is essential to note that all acquisition assays/acquisition inhibition assays in this study were repeated three times for consistency check.

### GST pull down

100 μg of both GST-AcrVA5 and GST proteins were incubated with 50 μl of settled Glutathione Agarose resin in 1 ml of GST binding buffer - comprised of 25 mM Tris (pH7.5), 1 mM EDTA, 0.01% NP-40, and 100 mM NaCl—at 4 °C for 30 min. After three washes, the pellet beads were resuspended in the GST binding buffer, and added 50 μg of Cas2. This mixture was then incubated overnight at 4 °C in sterile centrifuge tubes, making use of a gentle rocking motion in a rotating incubator to maintain an even suspension. Post incubation, the beads were washed carefully to preserve the interaction between Cas2 and AcrVA5 while eliminating non-specific proteins. Bead samples that were resolved in 1×SDS loading buffer were subsequently analyzed using SDS-PAGE and Coomassie staining methods.

### In vitro cleavage assay and cleavage inhibition assay

20 μM of Cas2/mutant protein were incubated with 1 μg of plasmid in a reaction buffer comprised of 20 mM Hepes (pH 7.5), 200 mM KCl and 2.5 mM MgCl$_2$ at 37 °C for a duration of 30 min. This did not apply to

the time gradient study where incubation times varied. For measuring pH-dependence, the same buffer was used, substituting Hepes with citrate, Tris, or CAPS. For assessing divalent ion-dependency, we replaced $Mg^{2+}$ in the reaction buffer with EDTA, $Ca^{2+}$, $Ni^{2+}$, $Mn^{2+}$, or $Cd^{2+}$. In the cleavage inhibition assay, to examine the inhibition of dsDNA cleavage by Cas2 mediated by AcrVA5, we first incubated the purified Cas2 protein with the AcrVA5 protein in a cleavage buffer composed of 20 mM Heps (pH 7.5), 200 mM KCl and 2.5 mM $MgCl_2$ at 4 °C for 30 min. This was followed by the addition of plasmids to the mixture, which was then further incubated at 37 °C for 30 min. We utilized molar ratios of Cas2 to AcrVA5 ranging from 1:0 to 1:4 in the assay, as depicted in Fig. 2c. To explore the AcrVA5-mediated inhibition of dsDNA cleavage by the mutants, the assay employed a Cas2 to AcrVA5 molar ratio of 1:2, with cleavage reactions conducted at 37 °C over a span of 20 min. The reaction process was halted using 0.4% SDS and 25 mM EDTA. Subsequently, Phenol-Chloroform extraction was executed to isolate the plasmids from Cas1-Cas2 and/or the AcrVA5 proteins. The DNA plasmids were then stained with a SYBR-safe DNA dye and run on a 1.5% agarose gel. Images were captured using the G-Box biomolecular imager (GE Healthcare).

### In vivo protospacer acquisition assays, and acquisition inhibition assays

The CRISPR locus was cloned into pTarget vector (Novagen) and transformed into *E. coli* BL21-AI to create *E. coli* BL21-AI-locus. Subsequently, Cas1 and Cas2 were cloned into pETDuet-1 vector (Novagen), while the protospacer and AcrVA5 were cloned into pACYCDuet-1 vector (Novagen). In the control group, BL21-AI-locus was transformed with pETDuet-1-Cas1+Cas2 and pACYCDuet-1-protospacer, whereas in the treatment group, the BL21-AI-locus was transformed with pETDuet-1-Cas1+Cas2 and pACYCDuet-1- protospacer-AcrVA5. All samples were cultured overnight in LB medium and then transferred (1:300) into 10 ml medium containing 0.2% L-arabinose, 0.1 mM IPTG, and antibiotics (50 µg/ml Spectinomycin, 50 µg/ml Ampicillin, and 25 µg/ml Chloramphenicol) to induce protein expression. The $OD_{600}$ of each culture was measured before transferring to 10 ml medium. After 20–24 h, the samples were spun down, and the sediment was dissolved in water and heated at 95 °C for 5 min. The treated samples were used as templates for PCR amplification of the CRISPR locus using specific primers. The amount of each sample used as PCR template was normalized. The PCR products were analyzed on 2% agarose gels. All reported results in this study were replicated three times.

### Isothermal titration calorimetry (ITC)

All ITC experiments were conducted at 25 °C using a MicroCal auto-ITC200 instrument. We filled the sample cell with 50 µM of Cas2 proteins in a buffer solution containing 20 mM Tris (pH 7.4) and 200 mM NaCl. The AcrVA5 concentration used was 400 µM, in the same buffer composition. We employed injection volumes of 0.4 µl each, with each injection lasting 0.8 s, and 90-s intervals between each injection. The heat absorbed during the interaction between AcrVA5 and Cas2 was measured directly, providing key information about the relative binding affinity (KD), stoichiometry (*n*), enthalpy (Δ*H*), and entropy (Δ*S*). For data analysis, we performed baseline correction, integration, and curve fitting using the independent binding modes available in the Sedfit software program.

### Peptide-scanning affinity capture

In the peptide-scanning affinity capture, biotinylated peptides were left to incubate with 50 µl of Streptavidin beads (Pierce) in 1 ml of binding buffer. This buffer contained 25 mM Tris (pH 7.5), 1 mM EDTA, 1% Triton X-100, and 100 mM NaCl, and the incubation occurred at 4 °C over a duration of 8 h. After washing the beads three times, we resuspended the pellet beads in the binding buffer and further incubated them with 50 µg of Cas2. This was done in sterile centrifuge

tubes, which were then placed at 4 °C overnight with a gentle rocking motion in a rotating incubator for even distribution. Subsequent to these stages, the beads were carefully washed to maintain the interaction between Cas2 and the biotinylated peptides. We then detached the bound biotinylated peptides and Cas2 from the beads using 50 µl of an elution buffer that was saturated with biotin. Following this, the samples were analyzed using a 6% Tris-Tricine-SDS gel and Coomassie staining.

### Crystallization, data collection and structure determination

The Cas2 protein was dialyzed against a buffer consisting of 20 mM Tris (pH 7.4), 150 mM NaCl, and 5% glycerol at 4 °C overnight. The Cas2 protein was then concentrated to approximately 12 mg/ml, while the peptides were diluted to a final concentration of 0.72 mM in Cas2, creating a Cas2 to peptide molar ratio of 1:1.2. This Cas2-peptide complex was used for crystallization trials using the hanging drop method in 24-well trays. After screening over 500 conditions, we were able to obtain well-diffracting crystals of the Cas2 and AcrVA5 peptide complex in buffers containing 15% PEG3350, 10% glycerol, and 0.1 M Tris (pH 6.4). Following optimization, we flash-froze single crystals using a cryo-solution, which included the crystallizing buffer plus 25% glycerol. Data collection took place at the Taiwan synchrotron radiation center and was processed using the HKL2000 software. Utilizing 5XVN as the search model, we solved the structure with MORDA/CCP4 and visualized it using the Coot software (www.ccp4.ac.uk). The data processing, as well as refinement statistic details of the structure, are listed in Table 1. Structural and electrostatics analysis were executed using the Chimera program (www.cgl.ucsf.edu/chimera/).

### Bio-dot Electrophoretic Mobility Shift Assays (EMSA)

Bio-dot EMSA experiments were conducted to evaluate the inhibitory effect of AcrVA5 on the protospacer (dsDNA) binding ability of the Cas1-Cas2 complex. The purified Cas1 and Cas2 proteins were individually dialyzed overnight at 4 °C against a buffer composed of 20 mM Heps (pH 7.4) and 150 mM NaCl. Post-dialysis, the proteins were combined in a 1:1 ratio and incubated for 30 min. This was followed by an additional overnight incubation with the inclusion of AcrVA5, also maintained at 4 °C. The purified Cas1-Cas2 and Cas1-Cas2-AcrVA5 proteins were then prepared in a buffer containing 20 mM HEPES (pH 7.4), 150 mM NaCl, 1 mM DTT, and 2 mM EDTA. These were subsequently coated on a Hybond-N+ membrane and immersed in a Blocking buffer (sourced from Thermo Fisher Scientific Inc., USA) for 30 min. Biotin-labeled protospacer was then added and incubated at 4 °C for 60 min, followed by UV cross-linking. Dot detection was carried out as per the instructions provided in the Light Shift Chemiluminescent EMSA kit for further analysis.

### Mass spectrometry

For in-gel digestion, co-expressed Cas2 protein bands with AcrVA5 and Cas2 bands expressed only were excised from the gel. After two washes, we added 100 µl of a decoloration buffer containing 50% acetonitrile and 25 mM ammonium bicarbonate. The samples were then reduced with 10 mM DTT and alkylated with 55 mM iodoacetamide. The dried gel pieces were rehydrated in 10 µl of 25 mM ammonium bicarbonate and 10% ACN, containing 0.1 µg of trypsin, to disintegrate the proteins into peptides. The tryptic peptides were then extracted twice using an extraction solution consisting of 67% acetonitrile and 5% trifluoroacetic acid. We combined the extracts and dried them in a SpeedVac dryer (Thermo Scientific). The dried tryptic peptides were re-dissolved in a buffer containing 0.1% formic acid and 5% acetonitrile and then desalted in a prepacked C18 column (100 µm × 3 cm, C18, 3 µm, 150 Å). The mobile phase A consisted of 0.1% formic acid and 5% acetonitrile, and the mobile phase B was composed of 0.1% formic acid and 95% acetonitrile. The peptides were separated using the Easy-nLC1200 system (Thermo Scientific) and

subsequently analyzed via MS (QEactive, Thermo Scientific). To ensure accuracy, all MS/MS spectra corresponding to acetylated peptides were manually examined.

## Statistics and reproducibility

No statistical method was used to predetermine sample size in this study. No data were excluded from the analyses. There was no requirement for randomization in our study.

## Reporting summary

Further information on research design is available in the Nature Portfolio Reporting Summary linked to this article.

## Data availability

The crystal structure data for the complex of Cas2 and AcrVA5-peptide have been deposited with accession codes 8IA4. The previously published PDB entries we have used can be found under accession codes 6IUF and 5XVN. Source data is included with this paper. Source data are provided with this paper.

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

## Acknowledgements

We would like to acknowledge the staff at Taiwan Synchrotron Radiation Resource Center (NSRRC) for assistance in data collection. We thank Ms Codrutza-Maria Dragu (Oxford University) for linguistic assistance. This work was supported by grant from National Natural Science Foundation in China (Grant No. 32071476 to X.M.).

## Author contributions

X.M. and M.B. conceived the study. M.B., W.S., J.L., and X.M. performed the experiments. X.M. supervised all of the research and wrote the manuscript. All authors read and approved the final manuscript.

## Competing interests

The authors declare no competing interests.
