## [Peer Review File · Nature Communications]

Insights into the inhibition of protospacer integration via direct interaction between Cas2 and AcrVA5Reviewer #1 (Remarks to the Author):

Anti-CRISPR proteins encoded by phages typically inactivate the CRISPR-Cas effectors through direct binding or enzymatic modifications. Cas1 and Cas2 are universally conserved among CRISPR systems and are responsible for the establishment of adaptive immunity, through the integration of new spacers into the CRISPR array. The study by Bi et al. tries to assign a new function to a previously defined anti-type V-A CRISPR protein, AcrVa5. This family has been shown to function as an acetylase, which inactivates Cas12a by acetylating a lysine residue in its PAM recognition site. Here the authors present evidence to argue that AcrVa5 also inhibits the spacer acquisition activity of Cas1-Cas2. If true, this will be the first example of an Acr targeting the spacer acquisition process, which would be quite a novel finding.

However, the manuscript in its current form has the following weaknesses that need to be addressed before this reviewer can be fully convinced. 1) The evidences presented by the authors could still be subject to alternative interpretations. 2) The authors seem to be hesitating between two mechanistic models, whether AcrVa5 inhibits Cas1-Cas2 interaction through binding and steric hindrance or it does so through the acetylation of the Cas1-Cas2 interface residue K55. The authors need to present evidence differentiating between these two models. 3) The entire study is built on biochemical analysis. In vivo functional analysis needs to be included to define the functional significance of the AcrVa5-Cas2 interaction in spacer acquisition. Here are the detailed questions for the authors to address.

1. Type II CRISPR system does not ubiquitously associate with type V-A system, therefore it is hard to imagine that a phage would evolve AcrVa5 to target both the type V effector Cas12a and the type II Cas2. It is more likely that this AcrVa5 is a broad spectrum Cas2 inhibitor (including V-A Cas2), or it has stronger inhibitory activity against the Cas2 in V-A system. Have the authors tested the binding activity of AcrVa5 against V-A Cas2s? Is the binding interface identified in this study conserved in V-A Cas2s? Is K55 conserved?

2. Page 8, 1st paragraph: ITC revealed the K_d of AcrVa5 for Cas2 is 9.39×10^{-5} mol/L. The author should convert that to standard molar concentrations. I am assuming it is 9.39×10^{-5} M, which is a very weak ~ 0.1 mM interaction. Granted that the Cas1-Cas2 interaction is also very weak, whether such a weak interaction between AcrVa5 and Cas2 could strongly impact spacer acquisition by Cas1 and Cas2 in vivo should be tested. The authors could electroporate annealed prespacer oligos into *E. coli* cells expressing their II-A Cas1-Cas2 and pCRISPR plasmid, in increasing concentrations of AcrVa5, to demonstrate the functional importance of this interaction.

3. Minor point: The binding stoichiometry was measured in ITC and presented in Figure 2B, however, the significance of this number is not discussed. Does the binding stoichiometry agree with the structure? What is the functional implication?

4. Figure 3A presents the crystal structure of Cas2 bound to an AcrVa5 peptide. A simulated annealing omit map should be presented for this AcrVa5 peptide in the supplemental figure to allow evaluation of the experimental map and occupancy. There should be discussions about the temperature factor and occupancy for this peptide.

5. AcrVa5 structure has been determined by other labs. Homology modeled structures or AlphaFold predicted structure of AcrVa5 should be aligned with the AcrVa5 peptide to generate a pseudo full complex structure. The authors can then discuss whether docking generates new insights or not. Are there major steric clashes? Is K55 in Cas2 accommodated at the active site of AcrVa5? Does it make sense for AcrVa5 to acetylate Cas2-K55?

6. Minor point – pg9, second paragraph – writing reads awkward in some places from here onwards. Consider to polish the writing further. For example "... suitable crystals of Cas2 in complex with AcrVA5 were not able to be obtained."

7. Minor point – pg 10, line 201: the mutagenesis data also confirms that lys55, a residue..." – K55 was not mentioned before. It was suddenly mentioned here.

8. Pg12: the evidence that Cas2-K55 is acetylated in a AcrVA5 dependent fashion is rather weak. Line 238 states "The MS results of the Cas2 proteins before and after being treated by ... (Figure 4A)". Only the MS data with AcrVa5 were presented in Figure 4A. The negative control is important for comparison but was not presented. If I interpreted correctly, K55 acetylation was not the predominant form of K55. If this is the case, could this mechanism be sufficient for the inhibition of Cas1-2 mediated spacer acquisition?

9. It would be better to present direct in vitro evidence that in the test tube, AcrVA5 can acetylate

K55 (or other lys) in Cas2 in acetyl-Co-A and NAD⁺ dependent manner. The author could then further define whether this modification or Cas2-binding is the driving force to inhibit spacer acquisition.

Reviewer #2 (Remarks to the Author):

The manuscript presents novel insights into the anti-CRISPR activities of AcrVA5, a protein previously known to function as an acetyltransferase that modifies a lysine residue in Cas12a, resulting in the loss of dsDNA cleavage activity. In this study, the authors demonstrate that AcrVA5 can also inhibit spacer acquisition in Type II CRISPR-Cas systems by acetylating a lysine residue, leading to the inhibition of the endonuclease activity of Cas2.

This study is particularly interesting as it reveals the ability of AcrVA5 to target different stages of CRISPR-Cas activity. The provided data is strong, although certain experiments lack necessary controls, and additional experiments are required to support some of the claims. Moreover, there is a need for a discussion of the findings in the context of previous results.

Main comments

1. Figure 3A does not clearly support the author's statement in the text regarding the interaction between AcrVA5-peptide and Cas2 mediated by the interaction between Lys59 of Cas2 and Ile3 of AcrVA5. Other residues, such as Lys55 and Ser56, appear to have similar interactions and are later shown to be more relevant, particularly Lys55, which is acetylated by AcrVA5. This should be clarified.

2. The authors claim that AcrVA5 causes the disassembly of Cas1-Cas2 complexes, but the data presented is insufficient to support this claim. To strengthen this conclusion, the authors should perform pull down assays with Cas1 and Cas2 proteins, with and without protospacer, in the presence and absence of AcrVA5. This would provide concrete evidence for the author's claim.

3. The proposed model suggests that the Cas1-Cas2-protospacer complex forms before AcrVA5 interacts with the Cas2 protein. However, there is no evidence presented for this order of events. It is possible that AcrVA5 interacts with Cas2 prior to complex formation. To address this, the authors should compare the inhibitory effect of AcrVA5 when pre-incubated with Cas2 before complex formation to AcrVA5 incubated after the formation of the Cas1-Cas2-protospacer complex. It would be valuable to determine if the complex can still form in the presence of AcrVA5.

4. In the discussion, the authors should refer to previous work on AcrVA5 (<https://pubmed.ncbi.nlm.nih.gov/30936526/>) which demonstrated its function as an acetyltransferase modifying a lysine residue in Cas12a. This Lys residue was crucial for PAM recognition, and its acetylation by AcrVA5 resulted in the complete loss of dsDNA-cleavage activity of Cas12a, similar to what the authors observed for Cas2. This mechanistic similarity is highly relevant and should be discussed in detail.

5. Several controls are missing in the provided figures. In Figure 2A, controls with proteins Cas2 and GST-AcrVA5 alone are absent. Figure 2C lacks controls with just dsDNA and dsDNA with only AcrVA5. Figure 4C is missing the control wt Cas2.

6. The steric hindrance mentioned in Figure 3B is not evident. Further clarification is needed.

Minor comments

1. Certain sections, such as the introduction and the beginning of the discussion, should be simplified and made more concise. The extensive explanation of CRISPR-Cas and its different types and subtypes in the introduction is excessive for the purpose of this paper. The concluding part of the introduction should be simplified to provide a better summary of the main findings. In the discussion, the introductory section is overly extensive, and lines 315-327 lack relevant discussion.

2. The methods section would benefit from language editing for clarity.

3. On line 143, the authors state that the free Cas1 and the protospacer-bound Cas1 do not interact with AcrVA5, but the data supporting this claim is not provided. The supplementary information should include this data.

4. Lines 154-157 should be rephrased to indicate that the Cas2 endonuclease activity has been demonstrated in several works (e.g. <https://pubmed.ncbi.nlm.nih.gov/26950513/> and <https://www.ncbi.nlm.nih.gov/pmc/articles/PMC5029534/>).

5. The order of panels in Figure S1 should be revised to match the order in which they are

mentioned in the text.

6. The titles provided at the bottom of each panel in all figures should be removed.

7. Figure 1B, "liner" should be corrected to "linear".

Reviewer #3 (Remarks to the Author):

Bi et al., describe the anti-CRISPR AcrVA5 (discovered as an inhibitor of the *Moraxella* Type VA CRISPR-Cas12a, MbCas12a) as a possible inhibitor of the Type II-A CRISPR-Cas1 and CRISPR-Cas2 integration complex. The authors assayed the ability of a panel of known Acrs to block the integration activity of Cas1-Cas2 in an in vitro plasmid integration assay revealing that AcrVA5 can prevent the integration activity. The authors go on to demonstrate a direct (but weak) association with the Cas2 protein and that incubation with AcrVA5 can block its endonuclease activity. Using a panel of peptides derived from AcrVA5, the authors go on to determine the X-ray structure of a peptide of AcrVA5 in complex with the Cas2 dimer revealing an interaction with a single copy of Cas2. Finally, the authors demonstrate that the previously published acetyltransferase activity of AcrVA5 is active against Cas2 by acetylating Lys55 in Cas2.

While this is the first reported example of an anti-CRISPR blocking the activity of Cas1-Cas2, it is the opinion of this reviewer that the data presented are incomplete and do not support the model presented. Further experiments are required to validate the role of AcrVA5 in blocking the Cas1-Cas2 integrase activity and to demonstrate that these findings are physiologically relevant to the field.

MAJOR COMMENTS

A key weakness of this study is the lack of in vivo data supporting the model proposed by the authors. The authors expressed AcrVA5 in cells with Cas2 and validated the acetylation of Lys55, but this was carried out in the absence of Cas1 making this physiologically irrelevant. The authors should repeat this experiment with Cas1-Cas2 and AcrVA5 present to validate the acetylation of Lys55 in Cas2. Following this, the authors can use the same experiment to assess the ability of AcrVA5 to block Cas1-Cas2 mediated acquisition in vivo which using in vivo spacer acquisition assays. This is a critical experiment to prove that AcrVA5 can (under physiologically relevant conditions) block the activity of Cas1-Cas2.

The authors have studied the ability of a type V anti-CRISPR to block a type II Cas1-Cas2 system. While it is possible that this type of crossover could be selected for, the authors provide little information or experimental evidence to support it. Is there evidence of AcrVA5 homologs in bacteria carrying the type II Cas1-Cas2 system? While Cas1 and Cas2 are conserved, they are subject to diversifying selection pressures that make them distinct across CRISPR types and subtypes. It is this reviewer's opinion that the authors need to test if AcrVA5 blocks type V Cas1-Cas2 integration activity in vivo.

The authors do not resolve the role of AcrVA5 directly binding to Cas2 from that of the AcrVA5-mediated acetylation of Cas2 at Lys55. The authors demonstrate that AcrVA5 can bind Cas2, but this interaction is weak (micromolar affinity as reported by ITC). A more relevant ITC experiment would be to assay AcrVA5 binding to the Cas1-Cas2 complex with and without protospacer present. As it stands it is unclear if the binding event described has any physiological relevance. The authors also present structural data showing a peptide interacting with Cas2 but do not relate this to the activity of the acetyltransferase domain. Is the structure representative of a catalytically active binding state where the acetyltransferase active site of AcrVA5 would be acetylating Lys55? Or is this binding event unrelated? The authors state steric hindrance would occur but present little data to support this statement. Could the authors model full-length AcrVA5 based off previously published structures and/or use AlphaFold to explore this?

The authors present incomplete data to support their hypothesis that AcrVA5 causes steric hindrance in the Cas1-Cas2 complex. The authors should validate that AcrVA5 can or cannot bind to a preformed Cas1-Cas2 complex with or without protospacer to support the hypothesis about steric hindrance based on the crystal structure. The authors should also assay the ability of Cas2

to function within the complex of Cas1-Cas2 post AcrVA5-mediated acetylation to validate the role of Lys55 acetylation in Cas1-Cas2 integration activity. The authors explored the role of Lys55 acetylation on Cas2 endonuclease activity, but this activity may not be relevant to Cas1-Cas2 integration activity in vivo or in vitro (both of which need to be explored).

The authors state that AcrVA5 causes partial dissociation of the protospacer and subsequently disassembly of the integration complex by EMSA (Figure 3C). However, the data do not justify this conclusion and the EMSA is flawed. Firstly, the intensity of the Cas1-Cas2-protospacer complex band is not affected by overnight incubation with AcrVA5 suggesting it has little to no effect on the integrity of the complex which contrasts with the authors' conclusions. Secondly, the concentration of the probe is too high for an EMSA to be meaningful. The concentration of the protospacer should be sub-stoichiometric relative to the Cas1-Cas2 complex. Finally, this assay needs three biological replicates and quantification to support the claims made by the authors that there is a difference observed.

MINOR COMMENTS

The authors state they tested 136 annotated anti-CRISPR proteins for the ability to block the integration activity of the Cas1-Cas2 complex but present only the data for AcrVA5. How were the other 135 inhibitors tested?

Could the authors comment on the identity of Band X within the integration assays? It appears as a result of the addition of protospacer and Cas1-Cas2 but it is not explained in the manuscript results.

Page 7, lines 137-139 – the authors state that the AcrVA5 protein does not significantly affect the integration complex Cas1 integrase activity. This doesn't make sense relative to the data presented which show that the AcrVA5 protein reduces the amount of Cas1-Cas2 mediated integration into supercoiled DNA. The authors should clarify this statement.

Page 8, lines 142-143 – these data should be shown as supplementary. Furthermore, how did the authors form a complex of Cas1 with protospacer in the absence of Cas2?

Could the authors explain why the X-ray crystal structure of the Cas2 dimer has only a single AcrVA5 peptide bound? The Cas2 dimer is symmetrical and should have the same binding site on the opposing face of the complex.

Page 10, line 199 – Could the authors elaborate on why 400Å² squared worth of interface would make it physiologically relevant?

The authors state "data not shown" for the AcrVA5 pulldown and these data should be included as supplementary.

REVIEWER COMMENTS

Reviewer #1 (Remarks to the Author):

Anti-CRISPR proteins encoded by phages typically inactivate the CRISPR-Cas effectors through direct binding or enzymatic modifications. Cas1 and Cas2 are universally conserved among CRISPR systems and are responsible for the establishment of adaptive immunity, through the integration of new spacers into the CRISPR array. The study by Bi et al. tries to assign a new function to a previously defined anti-type V-A CRISPR protein, AcrVA5. This family has been shown to function as an acetylase, which inactivates Cas12a by acetylating a lysine residue in its PAM recognition site. Here the authors present evidence to argue that AcrVA5 also inhibits the spacer acquisition activity of Cas1-Cas2. If true, this will be the first example of an Acr targeting the spacer acquisition process, which would be quite a novel finding.

However, the manuscript in its current form has the following weaknesses that need to be addressed before this reviewer can be fully convinced.

1) The evidence presented by the authors could still be subject to alternative interpretations.

Response: Thank you very much for your insightful feedback. We understand your concerns and fully acknowledge the potential for alternative interpretations. Although we have concluded based on our experimental results and the existing literature, we agree that there could be other interpretations. In the revised manuscript, we have expanded our discussion section to address the possible alternative interpretations of our data. We believe this will render our study more balanced and thorough.

Although AcrVA5 was known to function as an acetyltransferase, modifying a lysine residue in Cas12a during the interference step in the type V CRISPR-Cas system, *Mb*-AcrVA5 was also found to inhibit both the dsDNAse activity of *Mb*-Cas2 from type I-C and *Td*-Cas2 from type II-A (**Figure 2C and Figure S4**), as well as the integration process for both types, as reflected in **Figure 1E and Figure S1**.

2) The authors seem to be hesitating between two mechanistic models, whether AcrVA5 inhibits Cas1-Cas2 interaction through binding and steric hindrance, or it does so through the acetylation of the Cas1-Cas2 interface residue K55. The authors need to present evidence differentiating between these two models.

Response: Thank you for your comments. We are sorry for the confusion. In this study, we identified two new roles of AcrVA5:

- 1) *Mb*-AcrVA5 can inhibit the protospacer integration by the Cas1-Cas2 complex, primarily via binding and causing steric hindrance.
- 2) *Mb*-AcrVA5 has the ability to suppress the dsDNAse activity of both *Mb*-Cas2 from type I-C and *Td*-Cas2 from type II-A through acetylation of Lys⁵⁵.

Our biochemical and structural data indicate that AcrVA5 primarily inhibits integration by the Cas1-Cas2 complex via binding and ensuing steric hindrance. In an effort to test the protospacer binding capacity to the Cas1-Cas2 complex, we performed bio-dot electrophoretic mobility shift assays (EMSA) both in the absence and presence of AcrVA5. It was observed that when AcrVA5 was co-incubated with the integration complex (Cas1-Cas2-protospacer), the binding affinity between the Cas1-Cas2 complex and the protospacer was compromised (updated **Figure 3C**). Furthermore, the binding of AcrVA5 to the $\alpha 2$ helix of Cas2 creates steric hindrance with the adjacent protospacer (**Figure 3B**). Additionally, we conducted an integration assay to evaluate the impact of Lys⁵⁵ acetylation on the *in vitro* integration activity of Cas1-Cas2. It was observed that the Cas1-Cas2-protospacer complex, post-acetylation mediated by AcrVA5, demonstrated an integration capacity akin to the untreated Cas1-Cas2-protospacer complex, incorporating a similar amount of the protospacer into the plasmid (as depicted in **Figure S12**). As a result, acetylation modifications on Cas2 seem to produce a negligible effect on protospacer integration.

Moving to the second key role, our Mass Spectrometry (MS) results revealed that upon Cas2 protein treatment with AcrVA5, the amino acid lysine at position 55 (Lys⁵⁵) is likely the *in vivo* acetylation site of AcrVA5. In addition, creation of a K55A mutant resulted in Cas2 becoming resistant to inhibition by AcrVA5 when degrading dsDNA substrates. This suggests that the residue Lys⁵⁵ plays a pivotal role in mediating the suppression of Cas2 by AcrVA5 (**Figure 4C**). From these combined data, we propose that the suppression of Cas2's dsDNAse activity by AcrVA5 may predominantly occur via acetylation at the Lys⁵⁵ residue.

3) The entire study is built on biochemical analysis. *In vivo* functional analysis needs to be included to define the functional significance of the AcrVA5-Cas2 interaction in spacer acquisition.

Response: Thank you for your comments. Pursuant to the reviewer's suggestion, we performed *in vivo* protospacer acquisition and acquisition inhibition assays. These assays demonstrated that AcrVA5 could also inhibit the integration of the protospacer by the Cas1-Cas2 complex in a live setting. As depicted in **Figure 1F**, the efficacy of the spacer acquisition reaction was notably diminished in the presence of AcrVA5. These *in vivo* results align well with those from our earlier *in vitro* assay.

Here are the detailed questions for the authors to address.

1. Type II CRISPR system does not ubiquitously associate with type V-a system, therefore it is hard to imagine that a phage would evolve AcrVA5 to target both the type V effector Cas12a and the type II Cas2. It is more likely that this AcrVA5 is a broad spectrum Cas2 inhibitor (including V-A Cas2), or it has stronger inhibitory activity against the Cas2 in V-A system. Have the authors tested the binding activity of AcrVA5 against V-a Cas2? Is the binding interface identified in this study conserved in V-A Cas2s? Is K55 conserved?

Response: We appreciate your helpful commentary. In this study, the suppression exerted by *Mb*-AcrVA5 upon multiple Cas2 proteins was studied, namely: *Mb*-Cas2, part of the **type I-C CRISPR-Cas system**; *Mb*-Cas2, part of the type V-A CRISPR-Cas system; and *Td*-Cas2, part of the type II-A CRISPR-Cas system. Notably, *Mb*-AcrVA5 was able to inhibit the dsDNAse activity of both *Mb*-Cas2 and *Td*-Cas2. However, endonuclease activity within the *Mb*-Cas2 proteins from the type V-A CRISPR-Cas system was not noticed.

Despite rigorous experimentation, the incorporation of a spacer from the protospacer by the type V CRISPR-Cas system was not effectively observable. Conversely, for type I-C, the protospacer sequence was successfully integrated as a new spacer into the CRISPR array in the pCRISPR plasmid. The sequencing results affirm that the integrated spacer was correctly oriented at the leader-proximal end of the CRISPR array.

While AcrVA5 is established as an acetyltransferase, modifying a lysine residue in Cas12a during the interference step in the type V CRISPR-Cas system, it also demonstrated the inhibition of the dsDNAse activity of *Mb*-Cas2 from type I-C and *Td*-Cas2 from type II-A (**Figure 2C and Figure S4**), as well as the integration process for both types, as reflected in **Figure 1E and Figure S1**.

2. Page 8, 1st paragraph: ITC revealed the Kd of AcrVA5 for Cas2 is 9.39 E5 mol/L. The author should convert that to standard molar concentrations. I am assuming it is 9.39×10^{-5} M, which is a very weak ~0.1 nM interaction. Granted that the Cas1-Cas2 interaction is also very weak, whether such a weak interaction between AcrVA5 and Cas2 could strongly impact spacer acquisition by Cas1 and Cas2 *in vivo* should be tested. The authors could electroporate annealed prespacer oligos into *E. coli* cells expressing their II-A Cas1-Cas2 and pCRISPR plasmid, in increasing concentrations of AcrVA5, to demonstrate the functional importance of this interaction.

Response: Thank you for your comments. Our calculations indicate that the dissociation constant (Kd) between AcrVA5 and Cas2 is 1065 nM, implying a weak interaction between these proteins. In response to the suggestion, we conducted an *in vivo* protospacer integration assay where the Cas1-Cas2 complex recognized protospacers and integrated them into the pCRISPR plasmid within *E. coli* (as shown in **Figure 1F**). In line with our *in vitro* findings, AcrVA5 exhibited a significant suppressing effect on the protospacer integration during this *in vivo* assay.

3. Minor point: The binding stoichiometry was measured in ITC and presented in Figure 2B, however, the significance of this number is not discussed. Does the binding stoichiometry agree with the structure? What is the functional implication?

Response: Many thanks for your suggestions. In our study, the stoichiometry of Cas2 relative to AcrVA5, as derived from our ITC experiments, was approximately 0.596. These findings align with the structural observations from our high-resolution (2.07-Å) structure of the complex formed by a Cas2 dimer with a AcrVA5 peptide, highlighting the consistent one-to-two ratio between AcrVA5-peptide and Cas2 dimer.

4. Figure 3A presents the crystal structure of Cas2 bound to an AcrVa5 peptide. A simulated annealing omit map should be presented for this AcrVa5 peptide in the supplemental figure to allow evaluation of the experimental map and occupancy. There should be discussions about the temperature factor and occupancy for this peptide.

Response: Thank you for your comments. As recommended by the reviewer, we have produced an omit map visualized in **Figure S6**. In this respective structure, the model of the AcrVA5-peptide (with discernible and constructed residues: Ile-Glu-Leu-Ser-Gly) is represented as a ball and stick model. Correspondingly, the density maps are shown in analogous green. Simulated annealing omit maps were calculated utilizing Fo-Fc coefficients and phases derived from 40 cycles of simulated annealing computation of the model, excluding the AcrVA5-peptide. The contour level of the map was set at 2σ . In our high-resolution 2.07-Å structure, the average B-factors for chain-A (Cas2), chain-B (Cas2), and chain-Q (AcrVA5-peptide) were determined as 20.37, 18.44, and 29.14 Å², respectively. Heeding the reviewer's advice, the structural model was subjected to sequential refinements, including rigid-body, xyz, group B-factor, and individual B-factor adjustments, executed through Phenix.Refine. The model's geometry was observed using MolProbity, initially excluding the AcrVA5-peptide. The AcrVA5-peptide was subsequently included in the model, with the structure subjected to identical refinement procedures. These procedures included occupancy refinement, which began from an occupancy baseline of 0.5. Implementing this refinement approach was instigated by the identification that stabilizing occupancy at 1.0 within our structures resulted in considerable negative mFo-DFc electron density (-4σ to -6σ) surrounding the AcrVA5-peptide. This observation was consistently maintained, even when supplemental refinement procedures were accomplished with a settled occupancy of 1.0.

Hence, we concluded that the data-to-parameter ratio was sufficient to refine the B-factors and occupancy concurrently for the AcrVA5-peptide within our structure.

5. AcrVA5 structure has been determined by other labs. Homology modeled structures or alphafold predicted structure of AcrVA5 should be aligned with the AcrVA5 peptide to generate a pseudo full complex structure. The authors can then discuss whether docking generates new insights or not. Are there major steric clashes? Is K55 in Cas2 accommodated at the active site of AcrVa5? Does it make sense for AcrVA5 to acetylate Cas2-K55?

Response: Thank you for your comments. We superimposed our crystal structure with the pre-existing structures of the Cas1-Cas2-protospacer complex (PDB: 5XVN) and AcrVA5 (PDB: 6IUF). A pseudo full complex structure was produced and is now presented in the revised **Figure 3B**. This was achieved by aligning the AcrVA5-peptide and AcrVA5. Here, the central portion of the dual-forked protospacer occupies the concave region situated between two α -1 helices of the dimeric Cas2. The phosphate backbone of the protospacer establishes extensive contact with the positively charged residues located on the Cas2's α -1 helix, which protrudes into the major groove of the protospacer. In this hypothetical structure, the AcrVA5 associates with the lysine

residues within the α -2 helix, which is proximal to α -1 on Cas2. As depicted in **Figure 3B**, the binding of AcrVA5 to the α -2 helix of Cas2 induces steric conflict with the adjacent protospacer.

6. Minor point – pg9, second paragraph – writing reads awkward in some places from here onwards. Consider to polish the writing further. For example, “... suitable crystals of Cas2 in complex with AcrVA5 were not able to be obtained.”

Response: Thank you for your comments. As recommended, we have carefully revised and enhanced the language used throughout the updated manuscript.

7. Minor point – pg 10, line 201: the mutagenesis data also confirms that lys55, a residue...” – K55 was not mentioned before. It was suddenly mentioned here.

Response: Thank you for your comments. We sincerely apologize for the inaccuracies in our previous manuscript. In the revised manuscript, we have duly removed this sentence:

As shown later, the mutagenesis data also confirms that Lys⁵⁵, a residue in the α 2 helix of Cas2, is indeed crucial for the interaction between AcrVA5 and Cas2.

8. Pg12: the evidence that Cas2-K55 is acetylated in an AcrVA5 dependent fashion is rather weak. Line 238 states “The MS results of the Cas2 proteins before and after being treated by ... (Figure 4A)”. Only the MS data with AcrVa5 were presented in Figure 4A. The negative control is important for comparison but was not presented. If I interpreted correctly, K55 acetylation was not the predominant form of K55. If this is the case, could this mechanism be sufficient for the inhibition of Cas1-Cas2 mediated spacer acquisition?

Response: Thank you for your comments. In response to the reviewer's suggestion, we have included the Mass Spectrometry (MS) results of Cas2 protein prior to treatment with AcrVA5 in **Figure S9**. It clearly shows that Lys⁵⁵ was not acetylated before AcrVA5 treatment. In addition, creation of a K55A mutant resulted in Cas2 becoming resistant to inhibition by AcrVA5 when degrading dsDNA substrates. This suggests that the residue Lys⁵⁵ plays a pivotal role in mediating the suppression of Cas2 by AcrVA5 (**Figure 4C**). From these combined data, we propose that the suppression of Cas2's dsDNase activity by AcrVA5 may predominantly occur via acetylation at the Lys⁵⁵ residue.

In the updated manuscript, we present detailed findings on two specific points:

- 1) AcrVA5 has the ability to suppress the dsDNase activity of both *Mb*-Cas2 from type I-C and *Td*-Cas2 from type II-A though acetylation of Lys⁵⁵.
- 2) AcrVA5 can inhibit the protospacer integration by the Cas1-Cas2 complex, primarily via binding and causing steric hindrance.

Furthermore, an integration assay was carried out to ascertain the role of Lys⁵⁵ acetylation in Cas1-Cas2 integration activity *in vitro*. The results revealed that the amount of protospacer incorporated into the pCRISPR plasmid by the Cas1-Cas2-protospacer complex post-AcrVA5-mediated acetylation was quite similar to that of the

untreated Cas1-Cas2-protospacer complex (**Figure S12**). Therefore, we inferred that acetylation modifications of Cas2 exert minimal impact on protospacer integration by the Cas1-Cas2 complex.

9. It would be better to present direct *in vitro* evidence that in the test tube, AcrVA5 can acetylate K55 (or other Lys) in Cas2 in acetyl-Co-A and NAD⁺ dependent manner. The author could then further define whether this modification or Cas2-binding is the driving force to inhibit spacer acquisition.

Response: Thank you for your comments. Our results revealed that the acetylation modification on Cas2 has minimal effect on the protospacer's integration by Cas1-Cas2.

Reviewer #2 (Remarks to the Author):

The manuscript presents novel insights into the anti-CRISPR activities of AcrVA5, a protein previously known to function as an acetyltransferase that modifies a lysine residue in Cas12a, resulting in the loss of dsDNA cleavage activity. In this study, the authors demonstrate that AcrVA5 can also inhibit spacer acquisition in Type II CRISPR-Cas systems by acetylating a lysine residue, leading to the inhibition of the endonuclease activity of Cas2. This study is particularly interesting as it reveals the ability of AcrVA5 to target different stages of CRISPR-Cas activity. The provided data is strong, although certain experiments lack necessary controls, and additional experiments are required to support some of the claims. Moreover, there is a need for a discussion of the findings in the context of previous results.

Main comments

1. Figure 3A does not clearly support the author's statement in the text regarding the interaction between AcrVA5-peptide and Cas2 mediated by the interaction between Lys59 of Cas2 and Ile3 of AcrVA5. Other residues, such as Lys55 and Ser56, appear to have similar interactions and are later shown to be more relevant, particularly Lys55, which is acetylated by AcrVA5. This should be clarified.

Response: Thank you for your comments. The interaction between the AcrVA5-peptide and Cas2 is facilitated by a series of interactions: a hydrogen bond between Lys⁵⁹ of Cas2 and Ile³ of AcrVA5, as well as non-bond contacts between Lys⁵⁵ of Cas2 and Leu⁵ of AcrVA5, Gln⁵⁸ of Cas2 and Leu⁵ of AcrVA5, and Lys⁵⁹ of Cas2 and Ile³ of AcrVA5. The distances between these interacting atoms and a 2D representation of these binding interactions are detailed in **Figure S7**.

2. The authors claim that AcrVA5 causes the disassembly of Cas1-Cas2 complexes, but the data presented is insufficient to support this claim. To strengthen this conclusion, the authors should perform pull down assays with Cas1 and Cas2 proteins, with and without protospacer, in the presence and absence of AcrVA5. This would provide concrete evidence for the author's claim.

Response: Thank you for your comment. In our study, the dissociation constant (K_d) of the interaction between Cas2 and AcrVA5 was determined to be approximately 1065 nM. This is significantly higher than the K_d value for the Cas1-Cas2 interaction

reported ¹. A higher dissociation constant typically implies weaker binding affinity between the interacting partners. Therefore, it is improbable that AcrVA5 triggers the disassembly of the Cas1-Cas2 complexes. Instead, it appears to induce a partial disassembly between the Cas1-Cas2 and the protospacer. This is demonstrated in updated **Figure 3C**, where the presence of AcrVA5 resulted in reduced binding of the protospacer.

3. The proposed model suggests that the Cas1-Cas2-protospacer complex forms before AcrVA5 interacts with the Cas2 protein. However, there is no evidence presented for this order of events. It is possible that AcrVA5 interacts with Cas2 prior to complex formation. To address this, the authors should compare the inhibitory effect of AcrVA5 when pre-incubated with Cas2 before complex formation to AcrVA5 incubated after the formation of the Cas1-Cas2-protospacer complex. It would be valuable to determine if the complex can still form in the presence of AcrVA5.

Response: Thank you for the insightful comments. In literature, the dissociation constant (Kd) for the interaction between Cas1 and Cas2 is reported to be approximately 290 nM as measured by isothermal titration calorimetry ². However, our study found the dissociation constant (Kd) for the interaction between Cas2 and AcrVA5 to be significantly larger, at approximately 1065 nM. A smaller dissociation constant suggests a stronger binding affinity. Therefore, the binding between Cas1 and Cas2 is tighter compared to that between Cas2 and AcrVA5. Consequently, in our illustrated visualization model, in the Presence of AcrVA5, a Cas2 dimer recruits two dimeric Cas1 proteins to the leader sequence region, thereby forming a hexameric-dsDNA complex.

4. In the discussion, the authors should refer to previous work on AcrVA5 (<https://pubmed.ncbi.nlm.nih.gov/30936526/>) which demonstrated its function as an acetyltransferase modifying a lysine residue in Cas12a. This Lys residue was crucial for PAM recognition, and its acetylation by AcrVA5 resulted in the complete loss of dsDNA-cleavage activity of Cas12a, similar to what the authors observed for Cas2. This mechanistic similarity is highly relevant and should be discussed in detail.

Response: Thank you for the insightful comments. According to the literature, AcrVA5 is a type V anti-CRISPR protein found in the prophage regions of *Moraxella bovoculi* strains and functions as an N-acetyltransferase ³. Numerous cellular proteins, including Cas12a within the type V CRISPR-Cas system, can be acetylated by AcrVA5 ^{4, 5}. In this study, we examined the inhibitory effect of *Mb*-AcrVA5 on several proteins: *Mb*-Cas2 from the type I-C CRISPR-Cas system, *Mb*-Cas2 from type V-A CRISPR-Cas system, and *Td*-Cas2 from the type II-A CRISPR-Cas system. Notably, AcrVA5 was able to inhibit the dsDNase activity of both *Mb*-Cas2 and *Td*-Cas2. Although the endonuclease activity of the *Mb*-Cas2 proteins from the type V-A CRISPR-Cas system was observed, *Mb*-AcrVA5 acted as a broad-spectrum inhibitor for Cas2 across various species, including *Treponema denticola* and *Moraxella bovoculi* (**Figure 2C and Figure S4**). Based on these observations, we posited that AcrVA5 inhibits the dsDNA-

cleavage activity of both Cas12a (V-a) and Cas2 (II-A) using a similar mechanism, i.e., by acetylation of a lysine residue in the active site.

5. Several controls are missing in the provided figures. In Figure 2A, controls with proteins Cas2 and GST-AcrVA5 alone are absent. Figure 2C lacks controls with just dsDNA and dsDNA with only AcrVA5. Figure 4C is missing the control wt Cas2.

Response: Thank you for the comments. Per the suggestion, we have incorporated new figures into the revised manuscript.

6. The steric hindrance mentioned in Figure 3B is not evident. Further clarification is needed.

Response: Thank you for your comment. To elucidate the structural mechanisms of the AcrVA5-mediated inhibition of integration, our crystal structure was superimposed with previously determined structures of the Cas1-Cas2-protospacer complex (PDB: 5XVN) and AcrVA5 (PDB: 6IUF), which was achieved by aligning the AcrVA5-peptide and AcrVA5 (visualized in the updated **Figure 3B**). In this pseudo full complex structure, the central portion of the dual-forked protospacer is nestled into the concave area between the two $\alpha 1$ helices of the Cas2 dimer. The phosphate backbone of the protospacer interacts with the positively charged residues in the Cas2's α -1 helix, which itself extends into the major groove of the protospacer. Within this structured representation, the AcrVA5 binds to the lysine residues in $\alpha 2$, a region closely adjacent to $\alpha 1$ on Cas2. As displayed in **Figure 3B**, this binding of AcrVA5 to Cas2's $\alpha 2$ helix creates steric interference with the neighboring protospacer.

Minor comments

1. Certain sections, such as the introduction and the beginning of the discussion, should be simplified and made more concise. The extensive explanation of CRISPR-Cas and its different types and subtypes in the introduction is excessive for the purpose of this paper. The concluding part of the introduction should be simplified to provide a better summary of the main findings. In the discussion, the introductory section is overly extensive, and lines 315-327 lack relevant discussion.

Response: Thank you for your comments. We have modified the descriptions as suggested in the revised manuscript.

2. The methods section would benefit from language editing for clarity.

Response: Thank you for your comments. We appreciate your feedback. In response, we have thoroughly revised and enhanced the language used in the Methods section of our updated manuscript.

3. On line 143, the authors state that the free Cas1 and the protospacer-bound Cas1 do not interact with AcrVA5, but the data supporting this claim is not provided. The supplementary information should include this data.

Response: Thank you for your comments. In line with the recommendations from the reviewer, we conducted a GST pull-down assay, as demonstrated in **Figure S2**. The

results indicate that AcrVA5 does not interact with either the free Cas1 or the protospacer-bound Cas1.

4. Lines 154-157 should be rephrased to indicate that the Cas2 endonuclease activity has been demonstrated in several works (e.g. <https://pubmed.ncbi.nlm.nih.gov/26950513/> and <https://www.ncbi.nlm.nih.gov/pmc/articles/PMC5029534/>).

Response: Many thanks for your comments. We have modified the descriptions as suggested and the references have been added.

5. The order of panels in Figure S1 should be revised to match the order in which they are mentioned in the text.

Response: Thank you for the comments. We apologize for the mistakes we made. In the revised manuscript, order of panels in **Figure S1** (updated as **Figure S3**) has been changed in a professional way.

6. The titles provided at the bottom of each panel in all figures should be removed.

Response: Thank you for the comments. We have removed the titles at the bottom of each panel in all figures as suggested.

7. Figure 1B, “liner” should be corrected to “linear”.

Response: Thank you for the comments. We have modified the descriptions as suggested.

Reviewer #3 (Remarks to the Author):

Bi et al., describe the anti-CRISPR AcrVA5 (discovered as an inhibitor of the *Moraxella* Type VA CRISPR-Cas12a, MbCas12a) as a possible inhibitor of the Type II-A CRISPR-Cas1 and CRISPR-Cas2 integration complex. The authors assayed the ability of a panel of known Acrs to block the integration activity of Cas1-Cas2 in an in vitro plasmid integration assay revealing that AcrVA5 can prevent the integration activity. The authors go on to demonstrate a direct (but weak) association with the Cas2 protein and that incubation with AcrVA5 can block its endonuclease activity. Using a panel of peptides derived from AcrVA5, the authors go on to determine the X-ray structure of a peptide of AcrVA5 in complex with the Cas2 dimer revealing an interaction with a single copy of Cas2. Finally, the authors demonstrate that the previously published acetyltransferase activity of AcrVA5 is active against Cas2 by acetylating Lys55 in Cas2. While this is the first reported example of an anti-CRISPR blocking the activity of Cas1-Cas2, it is the opinion of this reviewer that the data presented are incomplete and do not support the model presented. Further experiments are required to validate the role of AcrVA5 in blocking the Cas1-Cas2 integrase activity and to demonstrate that these findings are physiologically relevant to the field.

MAJOR COMMENTS

1. A key weakness of this study is the lack of *in vivo* data supporting the model proposed

by the authors. The authors expressed AcrVA5 in cells with Cas2 and validated the acetylation of Lys55, but this was carried out in the absence of Cas1 making this physiologically irrelevant. The authors should repeat this experiment with Cas1-Cas2 and AcrVA5 present to validate the acetylation of Lys⁵⁵ in Cas2. Following this, the authors can use the same experiment to assess the ability of AcrVA5 to block Cas1-Cas2 mediated acquisition *in vivo* which using *in vivo* spacer acquisition assays. This is a critical experiment to prove that AcrVA5 can (under physiologically relevant conditions) block the activity of Cas1-Cas2.

Response: Thank you for the comments. Heeding the reviewer's recommendation, we performed *in-vivo* protospacer acquisition and acquisition inhibition assays. These tests validated that AcrVA5 can indeed hinder protospacer integration by the Cas1-Cas2 complex, also in a live setting. As demonstrated in **Figure 1F**, the presence of AcrVA5 greatly reduced the efficiency of the spacer acquisition reaction, a finding that aligns consistently with our earlier *in-vitro* assay results.

2. The authors have studied the ability of a type V anti-CRISPR to block a type II Cas1-Cas2 system. While it is possible that this type of crossover could be selected for, the authors provide little information or experimental evidence to support it. Is there evidence of AcrVA5 homologs in bacteria carrying the type II Cas1-Cas2 system? While Cas1 and Cas2 are conserved, they are subject to diversifying selection pressures that make them distinct across CRISPR types and subtypes. It is this reviewer's opinion that the authors need to test if AcrVA5 blocks type V Cas1-Cas2 integration activity *in vivo*.

Response: Thank you for the insightful comments. In this study, we examined the inhibitory effect of *Mb*-AcrVA5 on different proteins: *Mb*-Cas2 from the type I-C CRISPR-Cas system, *Mb*-Cas2 from the type V-A CRISPR-Cas system, and *Td*-Cas2 from the type II-A CRISPR-Cas system. Remarkably, *Mb*-AcrVA5 could inhibit the dsDNAse activity of both *Mb*-Cas2 and *Td*-Cas2. However, endonuclease activity within the *Mb*-Cas2 proteins from the type V-A CRISPR-Cas system was not noticed.

Despite comprehensive experimentation, we were unable to observe the effective incorporation of a spacer from the protospacer by the type V CRISPR-Cas system. In contrast, for type I-C, the protospacer sequence was effectively integrated as a new spacer into the CRISPR array in the pCRISPR plasmid. This process was highly efficient. The incorporated spacer was oriented correctly at the leader-proximal end of the CRISPR array, as confirmed by sequencing results.

Although AcrVA5 was known to function as an acetyltransferase, modifying a lysine residue in Cas12a during the interference step in the type V CRISPR-Cas system, AcrVA5 was also found to inhibit both the dsDNAse activity of *Mb*-Cas2 from type I-C and *Td*-Cas2 from type II-A (**Figure 2C** and **Figure S4**), as well as the integration process for both types, as reflected in **Figure 1E** and **Figure S1**.

3. The authors do not resolve the role of AcrVA5 directly binding to Cas2 from that of the AcrVA5-mediated acetylation of Cas2 at Lys⁵⁵. The authors demonstrate that AcrVA5 can bind Cas2, but this interaction is weak (micromolar affinity as reported by ITC). A more relevant ITC experiment would be to assay AcrVA5 binding to the Cas1-Cas2 complex with and without protospacer present. As it stands it is unclear if the binding event describe has any physiological relevance. The authors also present structural data showing a peptide interacting with Cas2 but do not relate this to the activity of the acetyltransferase domain. Is the structure representative of a catalytically active binding state where the acetyltransferase active site of AcrVA5 would be acetylating Lys⁵⁵? Or is this binding event unrelated? The authors state steric hinderance would occur but present little data to support this statement. Could the authors model full-length AcrVA5 based off previously published structures and/or use AlphaFold to explore this?

Response: Thank you for the insightful comments. As recommended, we carried out Isothermal Titration Calorimetry (ITC) to examine the binding of AcrVA5 to the Cas1-Cas2 complex, both with and without the presence of the protospacer. During ITC, the heat absorbed or released when AcrVA5 interacts with either the Cas1-Cas2 or Cas1-Cas2-protospacer complex is measured. However, the individual peaks can be challenging to integrate with software (shown as below).

Isothermal Titration Calorimetry of AcrVA5 Interaction with Cas1-Cas2 Complex, in the Presence and Absence of protospacer.

Our structural analysis illustrates how the interaction between the AcrVA5-peptide and Cas2 is mediated. More specifically, interactions occur between Lys⁵⁹ of Cas2 and Ile³ of AcrVA5 (via a hydrogen bond), and also between Lys⁵⁵ of Cas2 and Leu⁵ of AcrVA5, Gln⁵⁸ of Cas2 and Leu⁵ of AcrVA5, and Lys⁵⁹ of Cas2 and Ile³ of AcrVA5 (all involving non-bonded contacts). The distances between these contact atoms, as well as a 2D representation of the binding interface, are showcased in **Figure S7**. The structure therefore represents a catalytically active binding state wherein the acetyltransferase active site of AcrVA5 may acetylate Lys⁵⁵.

Furthermore, a pseudo full complex structure has been created by superimposition our crystal structure with the pre-existing structures of the Cas1-Cas2-protospacer complex (PDB: 5XVN) and AcrVA5 (PDB: 6IUF), as seen in the new **Figure 3B**. The central portion of the dual-forked protospacer is positioned in the concave region between the

two α 1 helices of the dimeric Cas2. It interacts with positively charged residues on Cas2's α 1 helix via its phosphate backbone, which neatly fits into the major groove of the protospacer. In this pseudo structure, the AcrVA5 binds to the lysine residues in Cas2's α 2 helix, which is situated adjacent to α 1. As depicted in **Figure 3B**, this binding of AcrVA5 to the α 2 helix of Cas2 creates steric hindrance with the neighboring protospacer.

4. The authors present incomplete data to support their hypothesis that AcrVA5 causes steric hindrance in the Cas1-Cas2 complex. The authors should validate that AcrVA5 can or cannot bind to a preformed Cas1-Cas2 complex with or without protospacer to support the hypothesis about steric hindrance based on the crystal structure. The authors should also assay the ability of Cas2 to function within the complex of Cas1-Cas2 post AcrVA5-mediated acetylation to validate the role of Lys⁵⁵ acetylation in Cas1-Cas2 integration activity. The authors explored the role of Lys⁵⁵ acetylation on Cas2 endonuclease activity, but this activity may not be relevant to Cas1-Cas2 integration activity *in vivo* or *in vitro* (both of which need to be explored).

Response: Thank you for the comments. To understand the structural mechanisms underpinning AcrVA5-mediated inhibition of integration, we superimposed our crystal structure with the pre-existing structures of the Cas1-Cas2-protospacer complex (PDB: 5XVN) and AcrVA5 (PDB: 6IUF). This was achieved by aligning the AcrVA5-peptide and AcrVA5. The resulting pseudo full complex structure is displayed in the newly generated **Figure 3B**. This diagram shows the centerpiece of the dual-forked protospacer infiltrating the concave region between the two α -1 helices of the Cas2 dimer, while the phosphate backbone of the protospacer interacts with the positively charged residues in Cas2's α -1 helix, extending into the major groove of the protospacer. Within this pseudo structure, AcrVA5 binds to the lysine residues in Cas2's α -2 helix, which lies close to the α -1 in Cas2. As depicted in **Figure 3B**, AcrVA5's binding to the α -2 helix of Cas2 results in steric hindrance with the adjacent protospacer.

As recommended, we performed an integration assay to determine the effect of Lys⁵⁵ acetylation on *in vitro* Cas1-Cas2 integration activity. Here, the post-AcrVA5-mediated acetylation Cas1-Cas2-protospacer complex incorporated a comparable amount of the protospacer into the plasmid to that of the untreated Cas1-Cas2-protospacer (**Figure S12**). Consequently, acetylation modifications on Cas2 appear to have a minimal effect on the integration of the protospacer.

In line with the reviewer's suggestion, we also conducted *in-vivo* protospacer acquisition and acquisition inhibition assays. These assays further validated that AcrVA5 can inhibit the protospacer's integration by the Cas1-Cas2 complex. As shown in **Figure 1F**, this spacer acquisition reaction was significantly hindered in the presence of AcrVA5, mirroring the findings from the *in-vitro* assay. In the *in-vivo* assays, Cas1, Cas2, and AcrVA5 were co-expressed, facilitating AcrVA5's acetylation of Cas2.

Thus, in the updated manuscript, we present detailed findings on two specific points:

- 1) *Mb*-AcrVA5 can inhibit the protospacer integration by the Cas1-Cas2 complex, primarily via binding and causing steric hindrance.
- 2) *Mb*-AcrVA5 could suppress the dsDNase activity of both *Mb*-Cas2 from type I-C and *Td*-Cas2 from type II-A, probably through acetylation of Lys⁵⁵.

5. The authors state that AcrVA5 causes partial dissociation of the protospacer and subsequently disassembly of the integration complex by EMSA (Figure 3C). However, the data do not justify this conclusion and the EMSA is flawed. Firstly, the intensity of the Cas1-Cas2-protospacer complex band is not affected by overnight incubation with AcrVA5 suggesting it has little to no effect on the integrity of the complex which contrasts with the authors' conclusions. Secondly, the concentration of the probe is too high for an EMSA to be meaningful. The concentration of the protospacer should be sub-stoichiometric relative to the Cas1-Cas2 complex. Finally, this assay needs three biological replicates and quantification to support the claims made by the authors that there is a difference observed.

Response: Response: Thank you for the comments. We sincerely regret the previous oversights. To rectify this, we conducted bio-dot EMSA experiments that were dependent on protein concentration to compare the binding affinities between protospacer and both Cas1-Cas2 and Cas1-Cas2-AcrVA5. As anticipated, Cas1-Cas2-AcrVA5 demonstrated a significantly lower binding affinity with protospacer compared to Cas1-Cas2. This suggests that the presence of AcrVA5 has notably reduced the binding affinity with protospacer (updated **Figure 3C**).

MINOR COMMENTS

1. The authors state they tested 136 annotated anti-CRISPR proteins for the ability to block the integration activity of the Cas1-Cas2 complex but present only the data for AcrVA5. How were the other 135 inhibitors tested?

Response: Thank you for the comments. Only a limited number of candidates, including *Mb*-AcrVA5, were found to inhibit protospacer integration following the pre-incubation of the integration complex.

2. Could the authors comment on the identity of Band X within the integration assays? It appears as a result of the addition of protospacer and Cas1-Cas2 but it is not explained in the manuscript results.

Response: Thank you for the comments. According to the literature, Band X is identified as pCRISPR topoisomers¹.

3. Page 7, lines 137-139 – the authors state that the AcrVA5 protein does not significantly affect the integration complex Cas1 integrase activity. This doesn't make sense relative to the data presented which show that the AcrVA5 protein reduces the amount of Cas1-Cas2 mediated integration into supercoiled DNA. The authors should clarify this statement.

Response: Thank you for the comments. We concur with the reviewer's observation that the statement ".....the AcrVA5 protein does not significantly affect the integration

complex Cas1 integrase activity." was unclear. To prevent any confusion, we have now removed this sentence from the manuscript.

4. Page 8, lines 142-143 – these data should be shown as supplementary. Furthermore, how did the authors form a complex of Cas1 with protospacer in the absence of Cas2?

Response: Thank you for the comments. In line with the reviewer's suggestion, we have included the results of the GST pull-down assay in **Figure S2**.

The Cas1 protein has the capacity to bind to double-stranded DNA on its own.

5. Could the authors explain why the X-ray crystal structure of the Cas2 dimer has only a single AcrVA5 peptide bound? The Cas2 dimer is symmetrical and should have the same binding site on the opposing face of the complex.

Response: Thank you for the comments. From our ITC experiments, the calculated stoichiometry of Cas2 to AcrVA5 was approximately 0.596. This aligns with our 2.07-Å resolution structure of the complex between Cas2 and the AcrVA5-peptide, which consists of one Cas2 dimer and one AcrVA5-peptide.

The observation that only a single peptide is bound to the Cas2 dimer in the crystal structure could be because of several reasons:

- 1) The AcrVA5-peptide might show differential binding affinity towards the binding sites, only favoring one over the other due to subtle differences in the orientation, flexibility, or the microenvironment of the binding site.
- 2) Due to flexibility or disorder, the electron density corresponding to the peptide on one site may not be well defined for proper modelling.

6. Page 10, line 199 – Could the authors elaborate on why 400 Å squared worth of interface would make it physiologically relevant?

Response: Thank you for the comments. Firstly, the interaction between Cas2 and the AcrVA5-peptide is mediated by nine non-bonded contacts and one hydrogen bond. Specifically, the hydrogen bond is between Lys⁵⁹ of Cas2 and Ile³ of AcrVA5, while the non-bonded contacts occur between Lys⁵⁵ of Cas2 and Leu⁵ of AcrVA5, Gln⁵⁸ of Cas2 and Leu⁵ of AcrVA5, and Lys⁵⁹ of Cas2 and Ile³ of AcrVA5. The distances between these atoms and a 2D representation of the binding interface are included in **Figure S7**.

Secondly, the Buried Surface Area (BSA) of the Cas2 and AcrVA5-peptide complex is 434.56 Å² per molecule, as listed in Table S2. This suggests the complex between Cas2 and the AcrVA5-peptide could have physiological relevance.

7. The authors state “data not shown” for the AcrVA5 pulldown and these data should be included as supplementary.

Response: Thank you for the comments. Following the reviewer's suggestion, we have included the results of the GST pull-down assay in **Figure S2** in the revised manuscript.

References

1. Nunez JK, Lee AS, Engelman A, Doudna JA. Integrase-mediated spacer acquisition during CRISPR-Cas adaptive immunity. *Nature* **519**, 193-198 (2015).
2. Nunez JK, Kranzusch PJ, Noeske J, Wright AV, Davies CW, Doudna JA. Cas1-Cas2 complex formation mediates spacer acquisition during CRISPR-Cas adaptive immunity. *Nature structural & molecular biology* **21**, 528-534 (2014).
3. Dong C, *et al.* Anti-CRISPRdb: a comprehensive online resource for anti-CRISPR proteins. *Nucleic acids research* **46**, D393-D398 (2018).
4. Dong L, *et al.* An anti-CRISPR protein disables type V Cas12a by acetylation. *Nature structural & molecular biology* **26**, 308-314 (2019).
5. Kang X, *et al.* Reversible regulation of Cas12a activities by AcrVA5-mediated acetylation and CobB-mediated deacetylation. *Cell Discov* **8**, 45 (2022).

Reviewer #1 (Remarks to the Author):

The authors addressed most of the reviewers' comments. The in vivo function data is quite helpful. However, some evidence remains quite weak. These include the weaker binding Kds, and the lack of a convincing complex structure between AcrVA5 and Cas1-Cas2, and the lack of demonstration that AcrVA5 directly inhibits its cognate type V Cas1-Cas2 acquisition machine. Because the concept is quite novel, I am willing to keep an open mind by agreeing to the publication of this work, on the condition that the reviewers' comments and rebuttals are published online as well.

The authors have made it clear that Cas2 K55 acetylation by AcrVA5 does not interfere with integration. They should therefore revise their model in figure 5 to remove Lys55 acetylation from panel D, because leaving this annotation on panel D will mislead the readers. They could include a separate panel clearly stating that Lys55 acetylation inhibits Cas2 nuclease instead.

Reviewer #2 (Remarks to the Author):

The manuscript has undergone substantial revisions by the authors, including the incorporation of several additional experiments that support their conclusions. The reviewer's comments have been duly addressed, leading me to recommend the publication of this work.

Reviewer #4 (Remarks to the Author):

This reviewer thinks that the manuscript should be further improved before acceptance by Nature Communications.

Major points

To validate the functional importance of the N-terminal region of AcrVA5 for Cas2 binding, the authors should perform functional assays, including (1) ITC, (2) in vitro inhibition assay, and (3) in vivo inhibition assay, using the L5A point mutant and the M1-G7 deletion mutant of AcrVA5.

P10L190: "Concurrently, we synthesized six peptides, each corresponding to different but equal parts of the AcrVA5 amino acid sequence."

- What does "different but equal parts" mean? This should be clarified.
- It would be better to add figure panels (in Fig. S5) showing the full-length AcrVA5 sequence, in which the sequences of the five peptides are highlighted, together with a predicted AcrVA5 structure, in which the five sequences are mapped.
- The authors should perform ITC experiments to determine the affinity of the five peptides to AcrVA5.

Minor points

P8L155: 1065 nM should be 1.1 μ M

P10L200: "The structure of the Cas2-peptide (3-IELSG-7) complex was determined by Phenix/Phaser.."

"...determined by Phenix/Phase" should be "...determined by molecular replacement".

P11L217: "Correspondingly, the density maps are shown in analogous green"
This sentence makes no sense and should be deleted.

P11L217: "Simulated annealing omit maps were calculated utilizing Fo-Fc coefficients and phases derived from 40 cycles of simulated annealing computation of the model, excluding the AcrVA5-peptide. The contour level of the map was set at 2σ ."
These sentences should be placed in the figure legend, rather than the main text.

P14L282: "...could be attributed to the acetylation of Lysine-acid"

What does "Lysine-acid" mean? Does it mean "a lysine residue"?

Fig. S6: The figure should be enlarged, and the key amino-acid residues should be labeled. In addition, a similar panel showing a $2F_o - F_c$ map should be presented as Fig. S6B.

REVIEWER COMMENTS

Reviewer #1 (Remarks to the Author):

The authors addressed most of the reviewers' comments. The *in vivo* function data is quite helpful. However, some evidence remains quite weak. These include the weaker binding *K_D*, and the lack of a convincing complex structure between AcrVA5 and Cas1-Cas2, and the lack of demonstration that AcrVA5 directly inhibits its cognate type V Cas1-Cas2 acquisition machine. Because the concept is quite novel, I am willing to keep an open mind by agreeing to the publication of this work, on the condition that the reviewers' comments and rebuttals are published online as well.

Response: Thank you for the comments.

The authors have made it clear that Cas2 K55 acetylation by AcrVA5 does not interfere with integration. They should therefore revise their model in figure 5 to remove Lys55 acetylation from panel D, because leaving this annotation on panel D will mislead the readers. They could include a separate panel clearly stating that Lys55 acetylation inhibits Cas2 nuclease instead.

Response: Thank you for the comments. We have modified the descriptions as suggested in the revised manuscript. We removed Lys55 acetylation from panel D in **Figure 5**.

Reviewer #2 (Remarks to the Author):

The manuscript has undergone substantial revisions by the authors, including the incorporation of several additional experiments that support their conclusions. The reviewer's comments have been duly addressed, leading me to recommend the publication of this work.

Response: Many thanks for your comments.

Reviewer #4 (Remarks to the Author):

This reviewer thinks that the manuscript should be further improved before acceptance by Nature Communications.

Major points

To validate the **functional importance** of the N-terminal region of AcrVA5 for Cas2 binding, the authors should perform functional assays, including (1) ITC, (2) *in vitro* inhibition assay, and (3) *in vivo* inhibition assay, using the **L5A** point mutant and the M1-G7 deletion mutant of AcrVA5.

Response: Thank you for your insightful comments. Based on your recommendations, we have conducted Isothermal Titration Calorimetry (ITC) to study the binding of Cas2 to the L5A point mutant and the M1-G7 deletion mutant of AcrVA5. In the ITC experiment, we measured the heat absorbed or released as Cas2 interacts with both the AcrVA5-L5A and M1-G7 deletion mutant of AcrVA5. Our findings revealed that the dissociation constant (*K_d*) of the interaction between Cas2 and AcrVA5-L5A was 1.2

Nm (**Figure S6F**). However, integrating the individual peaks for the M1-G7 deletion mutant of AcrVA5 with the software proved to be a challenge (**Figure S6G**).

In addition, we conducted both *in vitro* and *in vivo* integration assays to confirm the functional significance of the N-terminal region of AcrVA5 for Cas2 binding. During these procedures, adding AcrVA5-L5A to the Cas1-Cas2-protospacer led to less amount of the protospacer being incorporated into the plasmid compared to the untreated Cas1-Cas2-protospacer (**Figure S7A, S7B**). Conversely, the amount of protospacer incorporated into the pCRISPR plasmid by mixing the M1-G7 deletion mutant of AcrVA5 with the Cas1-Cas2-protospacer complex resembled that of the untreated Cas1-Cas2-protospacer complex (**Figure S7A, S7B**).

From these results, we deduce that AcrVA5-L5A interferes with the integration of the protospacer both *in vivo* and *in vitro*. However, the N-terminal truncated AcrVA5 appears to have a negligible effect on the protospacer integration, both *in vivo* and *in vitro*. These findings corroborate our previous *in vitro* and *in vivo* experiments, which vouched for the functional importance of the AcrVA5-Cas2 interaction in spacer acquisition.

P10-L190: "Concurrently, we synthesized six peptides, each corresponding to different but equal parts of the AcrVA5 amino acid sequence." What does "different but equal parts" mean? This should be clarified. It would be better to add figure panels (in **Fig. S5**) showing the full-length AcrVA5 sequence, in which the sequences of the five peptides are highlighted, together with a predicted AcrVA5 structure, in which the five sequences are mapped.

Response: Thank you for your comments. We have revised the manuscript as suggested, including modifying the descriptions. The complete sequence of AcrVA5 is depicted in **Figure S5C**, where the sequences of the five peptides are clearly highlighted. Furthermore, the corresponding sections of the peptides in the AcrVA5 structure are also identified and mapped in **Figure S5D**. These are labeled with the same colors as in the peptide sequence for easy correlation.

- The authors should perform ITC experiments to determine the affinity of the five peptides to AcrVA5.

Response: Thank you for your comments. Acting upon your recommendation, we conducted Isothermal Titration Calorimetry (ITC) to investigate the binding affinity of AcrVA5-peptides to Cas2, the results of which are presented in the **new Figure S6**. In ITC, we measure the heat absorbed or released during the interaction between the AcrVA5-peptide and Cas2. However, it appears that either the binding affinity is exceptionally weak, or there is no interaction occurring between Cas2 and the AcrVA5-peptide (**Figure S6A-6E**).

Minor points

P8-L155: 1065 nM should be 1.1 μ M

Response: Thank you for the comments. We have modified the descriptions as suggested in the revised manuscript.

P10-L200: “The structure of the Cas2-peptide (3-IELSG-7) complex was determined by Phenix/Phaser...”

“...determined by Phenix/Phase” should be “...determined by molecular replacement”.

Response: Thank you for the comments. We have modified the descriptions as suggested in the revised manuscript.

P11-L217: “Correspondingly, the density maps are shown in analogous green”

This sentence makes no sense and should be deleted.

Response: Thank you for the comments. We have modified the descriptions as suggested in the revised manuscript.

P11-L217: “Simulated annealing omit maps were calculated utilizing Fo-Fc coefficients and phases derived from 40 cycles of simulated annealing computation of the model, excluding the AcrVA5-peptide. The contour level of the map was set at 2σ .”

These sentences should be placed in the figure legend, rather than the main text.

Response: Thank you for the comments. We have modified the descriptions as suggested in the revised manuscript.

P14-L282: “...could be attributed to the acetylation of Lysine-acid”

What does “Lysine-acid” mean? Does it mean “a lysine residue”?

Response: Thank you for the comments. We are sorry for the confusion.

We have modified the descriptions as “...could be attributed to the acetylation of a lysine residue”

Fig. S6: The figure should be enlarged, and the key amino-acid residues should be labeled. In addition, a similar panel showing a $2F_o - F_c$ map should be presented as **Fig. S6B**.

Response: Thank you for your comments. We have amended the manuscript per your recommendations. In the revised version, new **Figure S8A** has been enlarged for clarity. We have labeled the key amino-acid residues, and a $2mF_o - DFC$ map has been included, which is now exhibited as new **Figure S8B**.

Reviewer #1 (Remarks to the Author):

I am satisfied with most of the responses from the authors. However, I am concerned about the ITC measurements and one specific response from the authors.

"Our findings revealed that the dissociation constant (K_d) of the interaction between Cas2 and AcrVA5-L5A was 1.2 nM (Figure S6F)."

I checked their Figure S6F, The heat trace does not show clear plateaus at either end. The K_d measurement would not be accurate as the result. The trace seems to suggest a micromolar level binding, rather than 1.2 nM. In fact, the annotation in the figure panel writes " $K: 8.11 \pm 3.51 \text{ E}5 \text{ M}^{-1}$ ".

In fact, their entire ITC measurements are of low quality. Large heat noises suggests they did not equilibrate the buffers in the chamber and the needle.

This entire assay needs to be re-done.

Reviewer #4 (Remarks to the Author):

Since the authors have addressed the reviewer's comments, I would like to recommend the publication of this work.

REVIEWER COMMENTS

Reviewer #1 (Remarks to the Author):

I am satisfied with most of the responses from the authors. However, I am concerned about the ITC measurements and one specific response from the authors.

"Our findings revealed that the dissociation constant (K_d) of the interaction between Cas2 and AcrVA5-L5A was 1.2 μM (Figure S6F)." I checked their Figure S6F, the heat trace does not show clear plateaus at either end. The K_d measurement would not be accurate as the result. The trace seems to suggest a micromolar level binding, rather than 1.2 nM. In fact, the annotation in the figure panel writes " $K: 8.11 \pm 3.51 \text{ E5 M}^{-1}$ ".
Response: Thank you for the comments. We re-conducted Isothermal Titration Calorimetry (ITC) experiments (Figure S6F). Based on our assessments, the dissociation constant (K_d) signifying the interaction between AcrVA5-L5A and Cas2 is calculated to be $\sim 1.3 \mu\text{M}$. This value suggests a relatively weak interaction between these two proteins.

In fact, their entire ITC measurements are of low quality. Large heat noises suggest they did not equilibrate the buffers in the chamber and the needle. This entire assay needs to be re-done.

Response: Thank you for the comments. Acting upon your recommendation, we re-conducted Isothermal Titration Calorimetry (ITC) experiments to investigate the interaction between AcrVA5-L5A/AcrVA5 peptides and Cas2 proteins. The results suggest that there might be minimal to no interaction between Cas2 and AcrVA5-peptides, pointing to a weak binding affinity. These findings are visually represented in Figures S6A-6E.

Reviewer #1 (Remarks to the Author):

I no longer have issues with the ITC data. I recommend the publication of this work.